# Modeling flexible behavior with remapping-based hippocampal sequence learning

Yoshiki Ito[1,2,3]*, Taro Toyoizumi[2,4]*

[1]Department of Neuroscience, Graduate School of Medicine, The University of Tokyo, Tokyo, Japan; [2]Laboratory for Neural Computation and Adaptation, RIKEN Center for Brain Science, Saitama, Japan; [3]Division of Visual Information Processing, National Institute for Physiological Sciences, Okazaki, Japan; [4]Department of Mathematical Informatics, Graduate School of Information Science and Technology, the University of Tokyo, Tokyo, Japan

## eLife Assessment

This manuscript reports a **valuable** modeling study on sequence generation in the hippocampus in a variety of behavioral contexts. The authors model context-depending decision making, and suggest that psychiatric disorders can be interpreted in terms of over or under representation of context information. The presentation is **solid**, and the work will interest the broad community of researchers studying cortical-hippocampal interactions and sequences.

*For correspondence:
yito@nips.ac.jp (YI);
taro.toyoizumi@riken.jp (TT)

Competing interest: The authors declare that no competing interests exist.

**Abstract** Animals flexibly change their behavior depending on context. It is reported that the hippocampus is one of the most prominent regions for contextual behaviors, and its sequential activity shows context dependency. However, how such context-dependent sequential activity is established through reorganization of neuronal activity (remapping) remains unclear. To better understand the formation of hippocampal activity and its contribution to context-dependent flexible behavior, we present a novel biologically plausible reinforcement learning model. In this model, Context selector promotes the formation of context-dependent sequential activity and allows for flexible switching of behavior in multiple contexts. This model reproduces a variety of findings from neural activity, optogenetic inactivation, human fMRI, and clinical research. Furthermore, our model predicts that imbalances in the ratio between sensory and contextual representations in Context selector account for schizophrenia and autism spectrum disorder-like behaviors.

## Introduction

Humans exhibit highly flexible behavior. However, a major challenge in solving various tasks with one neural network is that the same external stimulus can have different meanings depending on the context. For example, the word 'mouse' can mean either an animal or a PC device, depending on the context (*Figure 1A*). Therefore, for correct word recognition, the biological neural computation should not be based only on the word 'mouse' alone, but also on the context it appears in. In experiments, it is reported that the hippocampus is one of the most important regions for contextual behavior. Hippocampal neurons show sequential activity (*Buzsáki and Tingley, 2018*; *Skaggs and McNaughton, 1996*; *Wilson and McNaughton, 1993*) related to episodic memory (*Burgess et al., 2002*), the amount of reward (*Ambrose et al., 2016*), planning (*Ólafsdóttir et al., 2018*), and recall (*Carr et al., 2011*), and their representation depends on the context (*Hasselmo and Eichenbaum,*

2005; *Chen and Wilson, 2023*). Additionally, hippocampal neurons exhibit reorganized neural activity called remapping (*Bostock et al., 1991*; *Muller and Kubie, 1987*), which does not purely reflect the change in the external stimuli but task structure (*Jeffery et al., 2003*), and subjective context (*Sanders et al., 2020*). However, how context-dependent sequential activity in the hippocampus is established through remapping and how it contributes to flexible behavior remain to be understood.

Several theoretical models have been proposed to explain how hippocampal activity depends on context. The first approach uses the structure of the environment. The Tolman-Eichenbaum Machine (*Whittington et al., 2020*) and the Clone Structured Cognitive Graph (*George et al., 2021*) account for context-dependent neural activities, such as splitter cells (*Dudchenko and Wood, 2014*) and lap cells (*Sun et al., 2020*), by introducing graphical structure stored within the network. However, these models entail optimization procedures like backpropagation or the expectation-maximization algorithm (*Whittington et al., 2020*; *George et al., 2021*), which are not considered biologically plausible. The second approach uses eligibility trace to explain how past experiences, i.e., temporal context, are integrated into hippocampal activity (*Cone and Clopath, 2024*). In this framework, the length of the temporal context is constrained by the time constant of the eligibility trace. Neverthe-less, animals can flexibly estimate the current context using history of various lengths (*Barnett et al., 2014*), suggesting that hippocampal activity may not be bound by a fixed eligibility window. The third approach trains recurrent neural networks (RNNs) to replicate the dynamics of hippocampal activity (*Leibold, 2020*). While previous works have explored hippocampal sequential activity for planning (*Jensen et al., 2024*; *Mattar and Daw, 2018*; *Pettersen et al., 2024*; *Stachenfeld et al., 2017*) and hippocampal remapping for contextual inference (*Low et al., 2023*) separately, they have yet to elucidate how these two aspects jointly enable flexible behavior. A simple biologically plausible model-based reinforcement learning model that uses the Amari-Hopfield model for context selection and hippocampal sequences of various lengths as a state-transition model for long-horizon planning, relying on remapping driven by prediction errors to form state representation, would thus provide valuable insights into the neural mechanisms underpinning context-dependent flexible behavior.

We aim to understand how hippocampal remapping, driven by prediction errors, gives rise to the formation and use of context-dependent hippocampal sequences, providing a biologically plausible account of flexible behavior, including rodents and humans. Our key idea is as follows. When the external environment deviates from the expectations of the current subjective context, prediction errors arise and trigger remapping. This process recruits distinct subsets of neurons to encode novel experience, thereby establishing separate contextual memories and enabling flexible goal-oriented behavior in response to sudden environmental changes. To demonstrate the capability of this idea, we constructed a computational model comprising two modules: Context selector that selects the appropriate context based on prediction errors, and Sequence composer (hippocampus) that learns to compose neural activity sequences predicting future events by concatenating context-dependent hippocampal segments according to reward. Our model implements simple model-based reinforce-ment learning in ambiguous contexts, yielding flexible behavior using a biologically plausible synaptic plasticity rule. We show that it reproduces a range of context-dependent hippocampal activities, as well as the impairments associated with specific brain lesion studies.

Finally, our model predicts a relationship between deficits in model-based behavior and sensory processing. Clinical research has reported that patients with schizophrenia (SZ) or autism spec-trum disorder (ASD) often exhibit problems with both behavioral flexibility and sensory processing, including hyper- and hyposensitivity (*Javitt and Freedman, 2015*; *Watts et al., 2016*). These symp-toms frequently co-occur, but the underlying reason remains unclear. Our model shows that the relative sizes of the neural populations in the sensory-processing region and the context-processing region within Context selector are important for contextual inference, suggesting that treatments targeting sensory processing could improve cognitive flexibility in some psychoses.

## Results

As illustrated in *Figure 1B*, we modeled the neural mechanisms of context-dependent behavior as the interaction between two functional modules: Context selector ($X$), which selects appropriate contexts, and Sequence composer (hippocampus, $H$), which generates neural activity sequences that predict future events. We use the Amari-Hopfield network (*Amari, 1972*; *Hopfield, 1982*) with Hebbian plas-ticity for $X$. $X$ has two domains: a stimulus domain that represents external stimuli, and a contextual

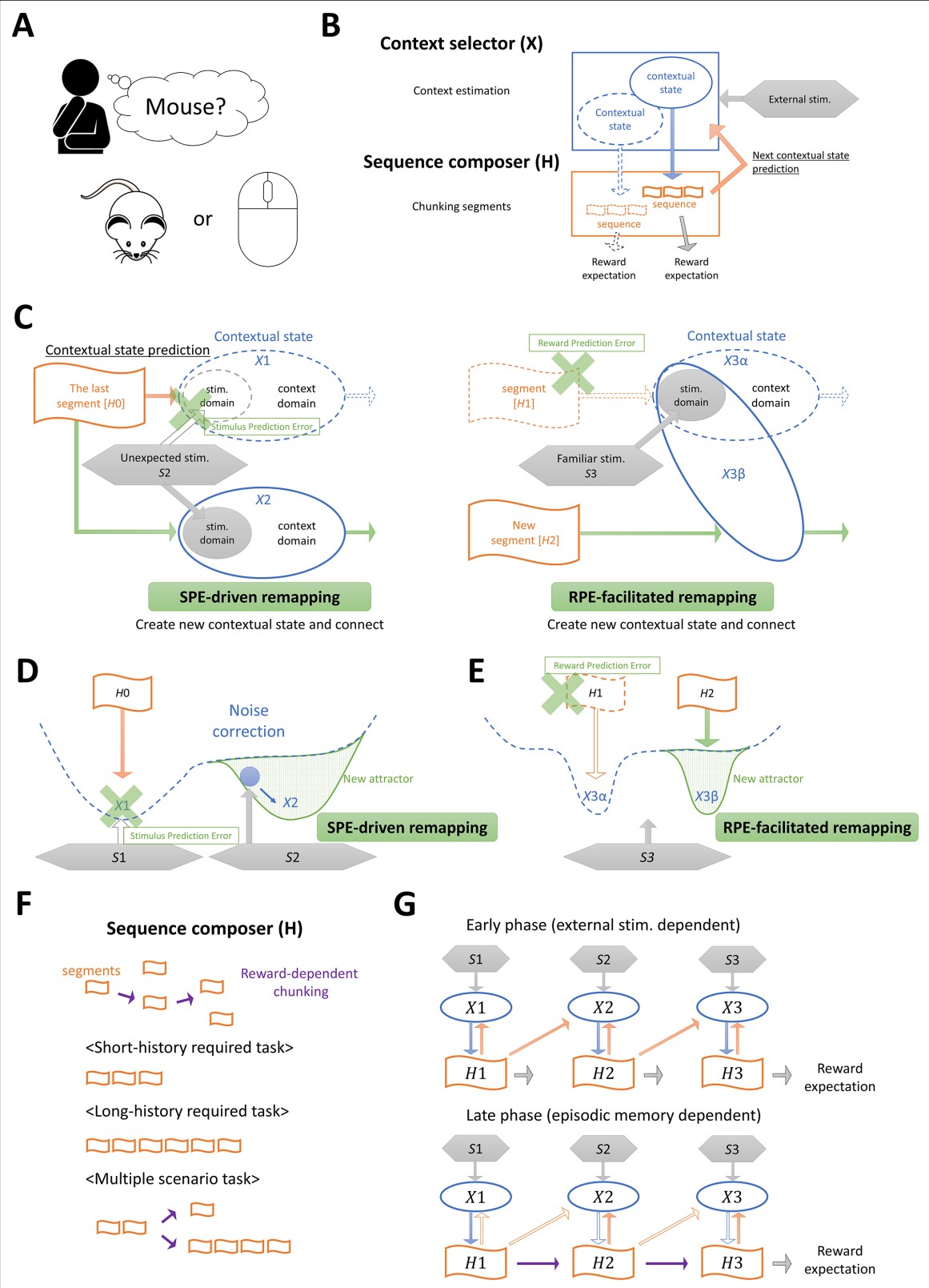

**Figure 1.** Schematic representation of our model. (**A**) An example of context-dependent cognition. Humans can understand the meaning of 'mouse' (an animal or a computer input device) depending on the context. (**B**) Our model involves two modules: Context selector (*X*) and Sequence composer (*H*). *X* chooses a context depending on the external stimuli and the input from *H*, and activates a sequence in *H*. This sequence is used for reward prediction. In addition, *H* sends predictive feedback about external stimuli to *X*. (**C**) The schematic figure of two kinds of remapping. Gray boxes indicate external

*Figure 1 continued on next page*

*Figure 1 continued*

stimuli, orange boxes indicate hippocampal segment (a part of hippocampal sequence), blue circles indicate contextual state, and green cross marks indicate the prediction error about external stimuli (left) and about reward (right). Solid lines indicate the actual state transition and dotted lines indicate virtual state transition that is created in the past transition. Green arrows indicate the synaptic potentiation related to remapping. (**D, E**) Attractor dynamics of Amari-Hopfield network related to sensory prediction error (SPE)-driven remapping (**D**) and reward prediction error (RPE)-facilitated remapping (**E**). Blue dotted lines indicate an energy landscape, and green solid lines indicate the chosen attractor as a result of remapping. (**F**) Hippocampal segments in *H* are combined depending on rewards (purple arrows) and formed into task-dependent sequences. Each sequence supports action planning and enables predictions of future external stimuli and rewards. (**G**) An example state transition related to hippocampal sequence formation. In the early phase, hippocampal neurons are activated through the input from *X*, while in the late phase, hippocampal neurons are activated through the recurrent input within *H*.

The online version of this article includes the following figure supplement(s) for figure 1:

**Figure supplement 1.** The algorithmic flowchart of the model.

domain that represents subjective contextual information. While the stimulus domain represents environmental states specified by the external stimuli, the contextual domain represents the contextual states for a given environmental states, which correspond to different subjective interpretations or associations of the external stimulus. *X* can stably store multiple contextual states by creating attractors in Amari-Hopfield model.

Our model's operations are algorithmic in nature indicated in *Figure 1—figure supplement 1*. When agents are at a starting point (i.e. a landmark), *X* initializes the neural activity of the contextual domain based on the external stimulus (see Materials and methods). When agents move to other environmental states, *X* receives predictive input from the lastly activated hippocampal segment together with the external stimulus and estimates the current context. Once *X*'s contextual state is set, it transmits the resulting output to *H*, which then activates an initial segment of *H*'s episodic sequence. *H* produces an episodic sequence corresponding to hippocampal replay (*Davidson et al., 2009*) or planning (*Ólafsdóttir et al., 2018*) based on its connectivity. For simplicity, we use a binary RNN for *H*, whose connectivity is updated by a three-factor Hebbian plasticity rule that depends on reward (see Materials and methods). Each replayed sequence is associated with actions (i.e. transition to the next environmental states) and two predictive outcomes: predicted future external stimuli and expected reward value. Based on the source of prediction errors, we consider two types of remapping: sensory prediction error (SPE)-driven remapping and reward prediction error (RPE)-facilitated remapping (*Figure 1C*). SPE-driven remapping is triggered when the mismatch between the predictive inputs from *H* to *X* and externally driven sensory inputs exceeds a threshold (see Materials and methods), causing *X* to either transition to a different contextual state or form a new one (*Figure 1D*). RPE-facilitated remapping is more likely to be triggered when the agents execute an action plan following a hippocampal sequence marked by a no-good indicator. The no-good indicator indicates that the action plan, i.e., the hippocampal sequence, has recently been associated with negative RPEs, possibly due to environmental changes (see Materials and methods). It then facilitates the exploration of alternative hippocampal sequences (*Figure 1E*). At the beginning of learning, hippocampal segments are not connected, and *H* yields only short sequences that generate immediate actions and short-term predictions. As learning continues, the three-factor Hebbian plasticity rule concatenates these segments, thereby creating longer sequences that reflect the task structure (*Figure 1F*). Thus, *H* learns to generate extended sequences that outline a course of actions and predict both reward and subsequent changes in the environment without explicit inputs from *X* (*Figure 1G*), forming a simple transition model for model-based reinforcement learning (*Coulom, 2007*). If a significant RPE error arises from a sequence, the agent explores a random action not specified by that sequence (see Materials and methods).

In the framework of reinforcement learning, our model can be mapped onto a Bayesian-adaptive model-based architecture in which contextual state serves as the root of Monte Carlo tree search (*Guez et al., 2013*) in a simple, largely stable environment with noiseless and unambiguous sensory stimuli, and only occasional abrupt changes. In this setup, prediction errors arise from the agent's lack of experience or due to abrupt environmental changes. Once a context selector *X* infers the hidden state, the sequence composer *H* generates episodic sequences that correspond to trajectories in a search tree, each branch representing possible action-outcome sequences. Just as Monte Carlo tree search explores potential future paths to evaluate expected rewards, *H* produces hippocampal

sequences that simulate future states and rewards based on its learned connectivity. In this way, $X$ defines the context that anchors the root of the tree, while $H$ expands the tree through replay or planning; thereby, our model provides a simplified algorithmic implementation of model-based reinforcement learning via tree search planning. However, these conceptual similarities are qualitative rather than quantitative. The goal of this work is not to achieve Bayesian optimality, but rather to show qualitative remapping-related processes that support goal-directed planning following epistemic errors.

## Splitter cells

Our model reproduces a range of hippocampal activity patterns that align with empirical data. First, we confirmed that our model reproduces the splitter cells reported in the hippocampus (*Dudchenko and Wood, 2014*). Splitter cells are a subset of hippocampal neurons that fire differentially on an overlapping segment of trajectories depending on where the animal came from, and/or where it is going. It is known that they do so based on information that is not present in sensory or motor patterns at the time of the splitting effect, but rather appear to reflect the recent past, upcoming future, and/or inferences about the state of the environment (*Duvelle et al., 2023*).

Experimentally, splitter cells are most often observed in an alternation task in a modified T-maze. Here, we simplified this task by using an environment with five discrete states ($S1$-$S5$), i.e., five discrete external stimuli (*Figure 2A*), where agents transition to the next state at each time step. In this environment, agents successfully solve this task by SPE-driven remapping, which creates different contextual states $X2\alpha$ and $X2\beta$ at an environmental state $S2$ based on where the agents came from, and thereby enabling context-specific exploration of which state to go ($S3$ or $S4$) (*Figure 2B*).

*Figure 2C* illustrates an example of both the environmental state transition and the corresponding contextual state transition of an agent, with each trial resetting upon visiting the reward sites ($S4$ or $S5$). The neural activity of $X$ at each contextual state is shown in *Figure 2D*, where the environmental states (e.g. $S1, S2, …$) are represented in the stimulus domain, and the contextual states (e.g. $X1$, $X2\alpha, …$) are represented in the context domain. A second contextual state at $S2$, $X2\beta$, was generated through SPE-driven remapping at the second visit of $S2$ (second trial) due to history mismatch between $S1 \rightarrow S2$ ($X1 \rightarrow X2\alpha$) and $S3 \rightarrow S2$ ($X3 \rightarrow X2\beta$) (see *Figure 1—figure supplement 1*). In Sequence composer, two types of neurons exist: state-coding neurons, which represent each contextual state, and transition-coding neurons, which encode transitions to successive contextual states given the contextual state indicated by the state-coding neuron (see Materials and methods). At each time step, one state-coding neuron and one transition-coding neuron are active in this order. Note that in the real brain, not only the hippocampus but also the premotor cortex and the basal ganglia contribute to action planning and execution (*Hikosaka et al., 2002*). Here, however, we focus on how simplified planning sequences are learned and composed in a context-dependent manner. In the example transition shown in *Figure 2C*, the agent selected an environmental state transition from $S2$ to $S4$ in the 2nd, 5th, and 8th trials, which corresponds to a contextual state transition from $X2\beta$ to $X4\beta$ in the $X$ module. However, because this transition was not rewarded, no synaptic potentiation occurred among hippocampal neurons. Subsequently, in the 11th trial, the agent attempted an environmental state transition from $S2$ to $S5$, which corresponds to the transition from $X2\beta$ to $X5\beta$ in the contextual states. The agent received a reward at $S5$, and the corresponding hippocampal sequence was strengthened, enabling the agent to acquire the alternation task in the following trials (*Figure 2E*).

In our model, most agents can solve this task (*Figure 2F*). As learning progresses, the length of hippocampal sequences increases, and eventually planning of the transition from one reward state to the next is possible (*Figure 2G*). Our model can be compared to the neural activity of the rats' splitter cells in the hippocampus during the modified T-maze task (*Wood et al., 2000*; *Figure 2H*). In our model, the transition-coding neurons exhibit right/left turn-specific firing at $S2$ after learning is complete (*Figure 2E and I*), replicating the emergence of splitter cells.

## Lap cells

The emergence of splitter cells explored above has also been studied in previous work (*Duvelle et al., 2023*; *Hasselmo and Eichenbaum, 2005*; *Katz et al., 2007*). However, these approaches generally assume that an appropriate temporal context—or a fixed length of sensory histories—is prepared in advance. This assumption becomes problematic in tasks where the number of required histories is unknown or changes dynamically: preparing too few histories results in failing to solve the tasks,

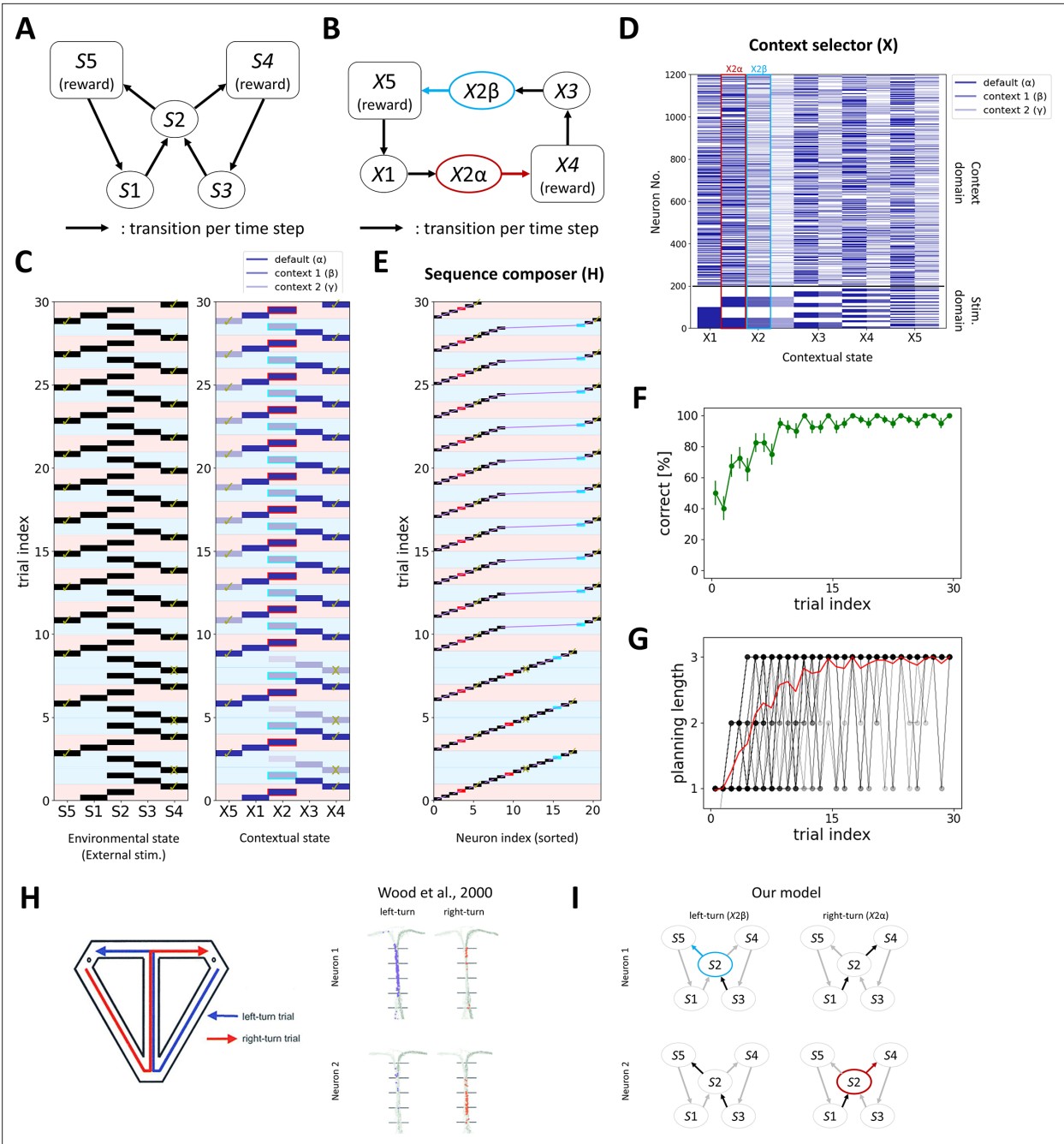

**Figure 2.** Our model replicates the emergence of splitter cells. (**A**) Simplified alternation task diagram. At each time step, the agents transition between environmental states. (**B**) A successful contextual state transition of our model. Preparing two different contextual states $X2\alpha$ and $X2\beta$ at $S2$ is necessary to solve this task. (**C**) An example environmental state transition (left) and contextual state transition (right). Check marks indicate the rewarded states, and cross marks indicate non-rewarded states. Red shades indicate the right-turn trials, and blue shades indicate the left-turn trials. (Right) The intensity of blue indicates the order of created contextual state, following history-driven remapping indicated in green triangles. Red outlines indicate $X2\alpha$, and blue outlines indicate $X2\beta$. (**D**) The corresponding neural activity of $X$ to each contextual state. The neurons in the stim. domain are sorted according to external stimuli. (**E**) The corresponding hippocampal activity at each contextual state. Red square indicates the transition-coding neuron of $S2$ to $S4$, and blue square indicates the transition-coding neuron of $S2$ to $S5$. Purple line indicates the hippocampal sequence, which is gradually lengthened in a reward-dependent manner. (**F**) The correct rate of our model. The error bars indicate the standard error of the mean ($N$=40). (**G**) The maximum number of environmental states ahead that the agents plan (planning length) gradually increases over learning. Black lines indicate the planning length of each agent, and the red line is their average. (**H**) Emergence of splitter cells in the hippocampus in the modified T-maze modification task (***Wood et al., 2000***). (**I**) The transition-coding neurons in our model replicate the emergence of splitter cells in $S2$.

while preparing too many slows down the search for a solution. Instead of preparing temporal context of fixed length in advance, our model uses remapping that adds new contextual states whenever a prediction error arises. This approach enables on-demand creation of contextual states and accelerates solution finding in dynamically changing tasks.

To show the advantage of our model, we demonstrate that our model replicates the emergence of lap cells (*Sun et al., 2020*). We set up a simplified discrete environment with a loop structure where the number of laps required to receive a reward varies (*Figure 3A*). Agents are initially rewarded for the shortest transitions through environmental states $S1 \rightarrow S2 \rightarrow S4$. After 20 trials, the environment changes, and the agents are rewarded for one lap transition, i.e., $S1 \rightarrow S2 \rightarrow S3 \rightarrow S2 \rightarrow S4$. It causes a large RPE (*no-good* indicator, see Materials and methods) and triggers RPE-facilitated remapping and exploration in the environment. During exploration, history mismatch triggers SPE-driven remapping in $S2$ and $S4$ as shown in *Figure 2*, and contextual states are discriminated into $X2\alpha$ / $X2\beta$ and $X4\alpha$ / $X4\beta$ based on the history (i.e. laps). In Sequence composer, the transition of contextual state $X1 \rightarrow X2\alpha \rightarrow X3\alpha \rightarrow X2\beta \rightarrow X4\beta$ is reinforced. After another 20 trials, the task environment changes again and the agents are rewarded for two laps, i.e., $S1 \rightarrow S2 \rightarrow S3 \rightarrow S2 \rightarrow S2 \rightarrow S3 \rightarrow S4$, or more. Either the shortest transition, $X1 \rightarrow X2\alpha \rightarrow X4\alpha$, or the 1-lap transition, $X1 \rightarrow X2\alpha \rightarrow X3\alpha \rightarrow X2\beta \rightarrow X4\beta$, is no longer rewarded, which triggers another RPE-facilitated remapping and exploration. During exploration, history mismatch occurs in $S2$, $S3$, and $S4$, and the contextual states for the second lap ($X2\gamma$, $X4\gamma$) are generated. Finally, the rewarded transition of contextual states and corresponding sequence, i.e., $X1 \rightarrow X2\alpha \rightarrow X3\alpha \rightarrow X2\beta \rightarrow X3\beta \rightarrow X2\gamma \rightarrow X4\gamma$, is reinforced (*Figure 3B*).

In our model, most agents can solve this task (*Figure 3C*). The episodic memory used for planning changes successfully depending on the environment (*Figure 3D*). This task is comparable with the 4-lap task for rats (*Sun et al., 2020*). In an environment where rats are rewarded for every four laps of a circuit, different hippocampal neurons fire for each lap. Our model replicates this result with the different hippocampal cells firing for different laps (*Figure 3E*). It is also reported that the inhibition of medial entorhinal cortex (EC) axons at CA1 attenuates the lap-specific activity (i.e. event-specific rate remapping [ESR]) without much affecting spatial encoding. Our model replicates this result by blocking the synaptic transmission from most of the neurons in the context domain of $X$ to $H$ (*Figure 3F*).

This task can also be solved by simply preparing temporal contexts with three steps of sensory history ($n=3$), which is the minimal number to solve this task (see Materials and methods for Model-free learning with temporal contexts). However, it takes much longer to find the correct transition for solving the 1-lap task than our model because it involves an excessive number of states (*Figure 3— figure supplement 1*). This result indicates that our model, which creates contextual states on demand, can perform better than the model with a fixed-length history.

To demonstrate the advantage of our model in a rapidly switching task that requires different history lengths, we show that an agent trained on both the 1-lap and 2-lap tasks can flexibly alternate between them in a reward-dependent manner (*Figure 3G*), selectively engaging hippocampal sequences of different lengths according to the current task context (*Figure 3H*). Together, these results illustrate how hippocampal lap-like representations emerge through learning and enable flexible context switching across tasks with distinct temporal demands.

## Planning in a stimulus-cued dynamic environment

In the real world, external stimuli dynamically change, and animals make plans and derive appropriate behavior by using the external stimulus as a clue. Here, we demonstrate that our model replicates key features of stimulus-related contextual behavior and its neural activity reported in experimental studies using SPE-driven remapping.

We consider a simplified environment of probabilistic cueing paradigm (*Ekman et al., 2022*). In this study, two auditory contextual cues probabilistically predicted distinct visual motion sequences, and fMRI decoding was used to examine the frequency of hippocampal replay. We simplified this task as shown in *Figure 4A*. In initial environment I, agents start from $S0$ and go to a state where one of two different external stimuli $S2$ or $S3$ is presented with different probabilities (p=0.8, 0.2, respectively). When $S2$ is presented, agents can get a reward at $S4$, whereas when $S3$ is presented, they can get a reward at $S5$. After 30 trials, the environment changes to II and the initial stimulus is switched to $S1$, not $S0$. In this environment, agents are rewarded at $S5$ and $S4$ when the external stimulus is $S2$ and $S3$, respectively (i.e. Reversal).

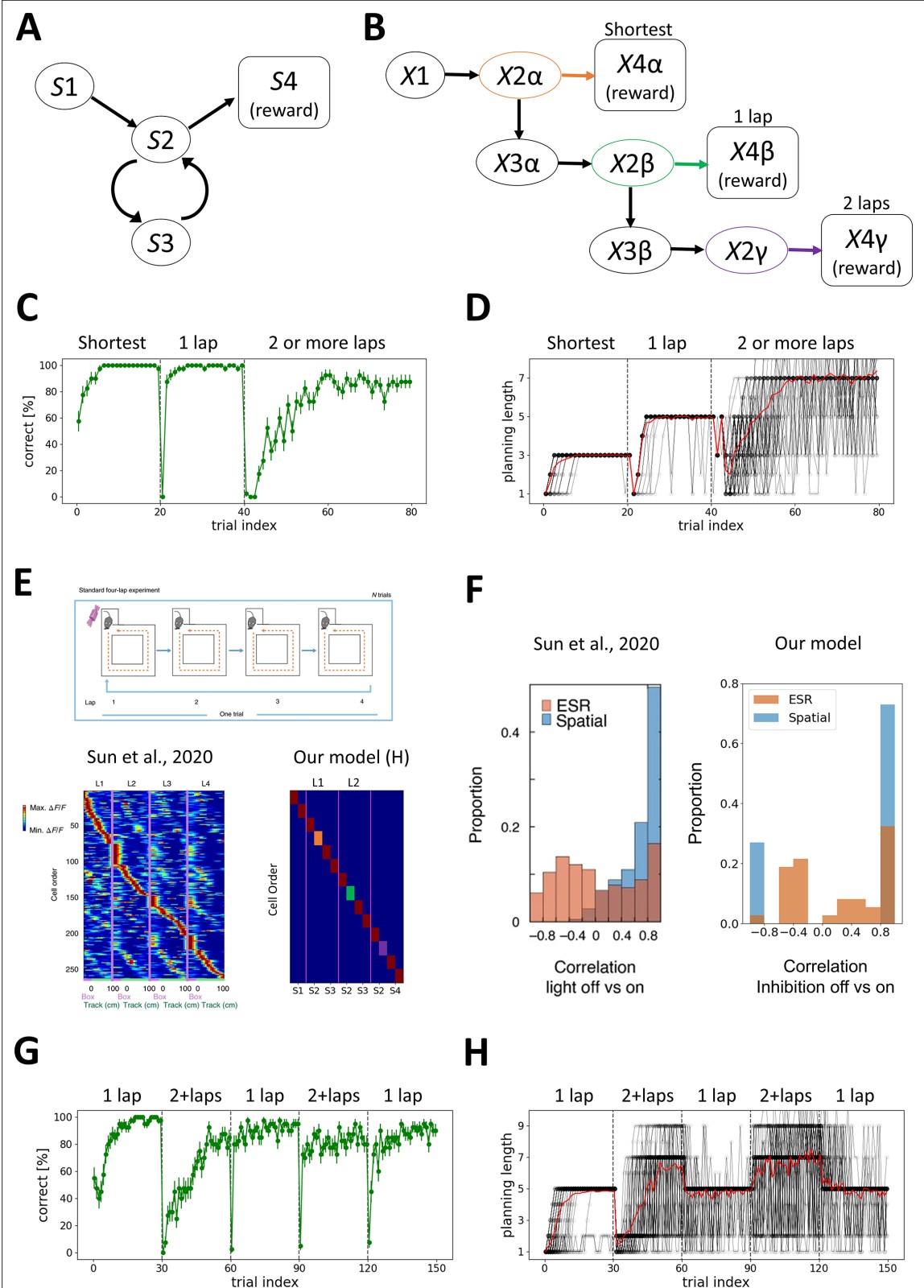

**Figure 3.** Our model replicates the emergence of lap cells. (**A**) Simplified 2-lap task diagram. Agents are rewarded for the shortest path ($S1 \to S2 \to S4$) for the initial 20 trials, for the 1-lap path ($S1 \to S2 \to S3 \to S2 \to S4$) for the next 20 trials, and for the 2 or more laps ($S1 \to S2 \to S3 \to S2 \to S3 \to S2 \to S4$, etc.) for the next 40 trials. (**B**) A successful contextual state-transition map of our model. The environmental states $S2$ and $S4$ are split into three contextual states $(X2\alpha, X2\beta, X2\gamma)$, $S3$ is split into two contextual states $(X3\alpha, X3\beta)$, and $S4$ is split into three contextual states $(X4\alpha, X4\beta, X4\gamma)$. (**C**) The correct rate

*Figure 3 continued on next page*

*Figure 3 continued*

of our model. The error bars indicate the standard error of the mean ($N$=40). (**D**) The planning length gradually increases during learning, depending on the task demand. The black lines indicate the planning length of each agent, and the red line is their average. (**E**) The comparison of (left) lap cells in the hippocampus in the 4-lap task (*Sun et al., 2020*) and (right) our results of active neurons in $H$ module. The $x$-axis represents each time step (corresponding to environmental states), and the $y$-axis shows the sorted activity of $H$ module. The transition-coding neurons at $S2$ in 2-lap task are indicated in orange, and green and purple squares corresponding to (**B**). (**F**) The inhibition experiment of medial entorhinal cortex axons at CA1. Event-specific rate remapping (ESR) cells show a weak lap-specific correlation (ESR correlation) between light-on trials and light-off trials, while they show a strong spatial correlation between light-on trials and light-off trials (left). Our model replicates the result qualitatively with the inhibition on and off (right). (**G**) The correct rate of 1-lap and 2-or-more-lap alternation task. The error bar indicates the standard error of the mean ($N$=40). (**H**) The planning length adapts flexibly to the task demand.

The online version of this article includes the following figure supplement(s) for figure 3:

**Figure supplement 1.** 2-Lap task with model-free learning with temporal contextual states.

In such a stochastic environment, the agents need to switch transition rules according to the external stimuli regardless of the prediction about the external stimuli beforehand. SPE-driven remapping (*Figure 1D*) enables our model to quickly change or generate the different context when the prediction error about the external stimuli occurs. For instance, in environment I, two rewarded contextual transitions exist: a more likely one ($X0 \rightarrow X2\alpha \rightarrow X4\alpha$) and a less likely one ($X0 \rightarrow X3\alpha \rightarrow X5\beta$) (*Figure 4B*). When an agent predicts the major stimuli ($S2$) at the initial state ($S0$) but minor stimuli ($S3$) are presented, the agent stops the sequence-based action loop (*Figure 1—figure supplement 1*), and SPE-driven remapping occurs, which switches the contextual state from $X2\alpha$ to $X3\alpha$ and the corresponding hippocampal sequence. As a result, the agents choose the correct transition regardless of prior prediction (*Figure 4B*).

In our model, most agents can learn to make appropriate transitions depending on the external stimuli. Importantly, they show a one-shot switch between environments I and II when the agents experience the environment for the second time (*Figure 4C*). This is because contextual states for $S2$ and $S3$ are generated differently in environments I and II, i.e., $X2\alpha, X3\alpha$ for environment I and $X2\beta, X3\beta$ for environment II, through SPE-driven remapping. The length of the planning sequence used in the actual transition converges to between 2 and 3 because agents reselect the hippocampal sequence and the contextual state when the external stimuli differ from predictions and SPE-driven remapping is triggered (*Figure 4D*). The probability of predicted external stimuli ($S2$ or $S3$) based on the generated sequences matches with the actual probability (p=0.8, 0.2, respectively) (*Figure 4E*), because of the reward-dependent synaptic plasticity in the hippocampus (see Materials and methods). This result replicates *Ekman et al., 2022*, who showed that the probability of the contextual cues is reflected in the statistically significant differences in hippocampal replay probability in humans (*Figure 4F*).

Our model is applicable to context selection under ambiguous external stimuli. *Julian and Doeller, 2021*, used a similar task structure as *Figure 4A* in humans and reported that the contextual representations and realignment in the hippocampus under ambiguous external stimuli predict context-dependent behavior. In the training phase, agents are put into either Square (Sq) or Circle (Ci) virtual reality arena, and then one of two target objects ($S2$ or $S3$) is randomly specified by word with equal probability. Depending on the arena type, the agents decide to transit to $S4$ or $S5$ to get a reward. In the test phase, subjects are put into either Sq, Ci, or their morphed version, Squircle (SC) arena, i.e., mean value of Sq and Ci. Under SC arena, the agents transit depending on the subjective context of either Sq or Ci. Note that reward feedback is not given in the test phase (*Figure 4G*).

Our model successfully learns this task, and the agents show context-dependent behaviors under Sq or Ci arena in the test phase (*Figure 4H*). Additionally, our model replicates the experimental results of SC as the mixed Sq- or Ci-like behaviors (*Figure 4I*). In humans, the Sq- or Ci-like behaviors are well decoded in the hippocampus, but it degrades under SC condition (*Julian and Doeller, 2021*). Our model replicates this result with a degraded decoding score under SC condition (*Figure 4J*). Here, three reconstruction cases are observed in $X$ under SC condition: Sq context reconstruction, Ci context reconstruction, and a default context usage of SC due to $X$'s failure to converge (see Materials and methods). In the last case, the agents make a random transition by recruiting new hippocampal neurons. Therefore, behavioral decoding based on hippocampal neural activity is lower than that under the Sq and Ci conditions (*Figure 4J*). This result is consistent with the findings of *Julian and Doeller, 2021*.

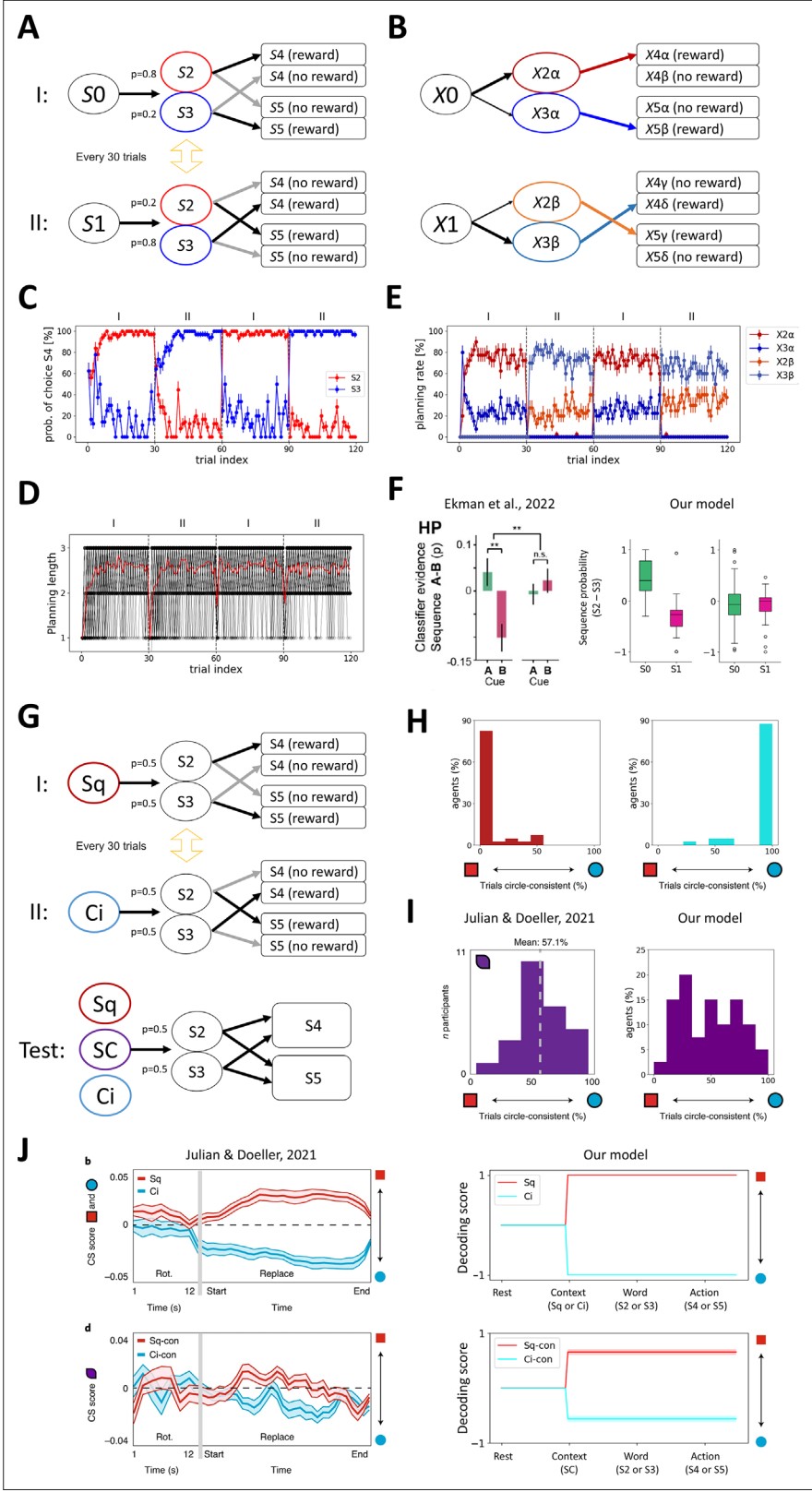

**Figure 4.** Our model replicates key features of human neural activity in dynamic environments. (**A**) Simplified probabilistic cueing task diagram. In environment I, agents start at *S0* and move to *S2* or *S3* randomly (*S2* for p=0.8 and *S3* for p=0.2) and receive a reward in *S4* when they come from *S2* and in *S5* otherwise. In environment II, agents start at *S1* and move to *S2* or *S3* randomly (*S2* for p=0.2 and *S3* for p=0.8) and receive a reward in *S5* when

*Figure 4 continued on next page*

eLife Research article

Neuroscience

*Figure 4 continued*

they come from *S2* and in *S4* otherwise. The environment switches between the two every 30 trials. (**B**) A successful context map of this task. *S2* and *S3* are split into two contextual states, and *S4* and *S5* are split into four contextual states. The hippocampal connections are built for rewarded conditions only. (**C**) The probability of choosing *S4*. The red/blue line shows its mean when *S2*/*S3* is presented. The error bars indicate the standard error of the mean (*N*=40). (**D**) The planning length gradually increases over learning and converges to 3. The black lines indicate each agent's planning length, and the red line is their average. (**E**) The probability of generating a specific planning sequence at *S0* or *S1*. The expected states (*S2* or *S3*) are modulated according to the environment. (**F**) Our model behavior is similar to the human fMRI result of the cue-probability-dependent hippocampal replay (*Ekman et al., 2022*). Paired sample t-test. **p<0.01. (**G**) Simplified task diagram (*Julian and Doeller, 2021*). The training phase is the same as (**A**), but the contextual stimuli of Square (Sq) or Circle (Ci) are initially presented, and the probability of *S2* and *S3* is equal. In the test phase, either one of Sq, Ci, or the mixture stimuli of Sq and Ci (Squircle: SC) are presented, and the agent transfers following their faith. Reward feedback is not given in the test phase. (**H**) The transition probability under Sq context (left) and Ci context (right). (**I**) The transition probability under SC context of the human patients in *Julian and Doeller, 2021* (left) and our model (right). (**J**) Comparison of behavioral decoding accuracy from hippocampal fMRI activity of *Julian and Doeller, 2021* (left) and hippocampal neural activity of our model (N=40) (right). The x-axis represents each time step (corresponding to environmental states), and the y-axis shows the decoding accuracy of each context based on hippocampal activity. The error bars indicate the standard error of the mean. Our model replicates the worse decoding accuracy in SC context (bottom) than Sq or Ci context (top).

## Prediction related to sensory processing and flexible behavior

Our model does not only replicate a variety of experimental results but also makes predictions. In clinical research, it has been reported that issues related to behavioral flexibility and sensory processing often co-occur in certain psychiatric conditions, including SZ (*Javitt and Freedman, 2015*) and ASD (*Watts et al., 2016*). Many studies have reported that both symptoms are linked to the dysfunction of the prefrontal cortex (PFC) (*Kaplan et al., 2016*; *Watanabe et al., 2012*); however, the reasons for their co-occurrence are not yet fully understood.

We assume that this dysfunction corresponds to hypo-/hyper-representation of stimulus information in *X*. To investigate this hypothesis, we altered the ratio of neurons in the context domain and sensory domain in *X* in our model. We used the same task described in *Figure 4A* with equal probability transitions to *S2* and *S3* (*Figure 5A*). When the stimulus domain is relatively underrepresented, the reconstruction of contextual state in the Amari-Hopfield network tends to infer contextual states based on the context domain rather than the stimulus domain. Consequently, it converges to an incorrect attractor that is not assigned to the current environmental state, thereby increasing perceptual error for external stimuli (hallucination-like effects). Moreover, SPE-driven remapping and the corresponding synaptic plasticity occur more frequently. In contrast, when the stimulus domain is overrepresented, the Amari-Hopfield network rarely assigns multiple contextual states to a given environmental state, leading to an overuse of default contextual states (see *Figure 5B* and Materials and methods).

Consistent with this prediction, when the stimulus domain is relatively underrepresented, agents fail to rapidly switch to the second experience in environments I and II (*Figure 5C*). This failure is accompanied by an increased probability of context selections that differ from the true environmental state (hallucination-like effects). Moreover, the hallucination-like effects increase SPE-driven remapping, which occasionally leads to overlaps in context allocation in *H* (see Materials and methods), thereby accelerating the frequency of hallucination-like effects and leading to a decline in task performance. In contrast, when the stimulus domain is relatively overrepresented, persistent behavior is observed, and the correct rate in environment II becomes lower than in environment I (*Figure 5C*). This is accompanied by an increased probability of default context usage due to failures in contextual state reconstruction (see Materials and methods) in environment II. Thus, our model predicts a relationship between sensory processing and behavioral flexibility in some psychosis.

## Discussion

In this study, we proposed a simple, model-based reinforcement learning model equipped with two functional modules: Context selector and Sequence composer. We introduced two kinds of prediction

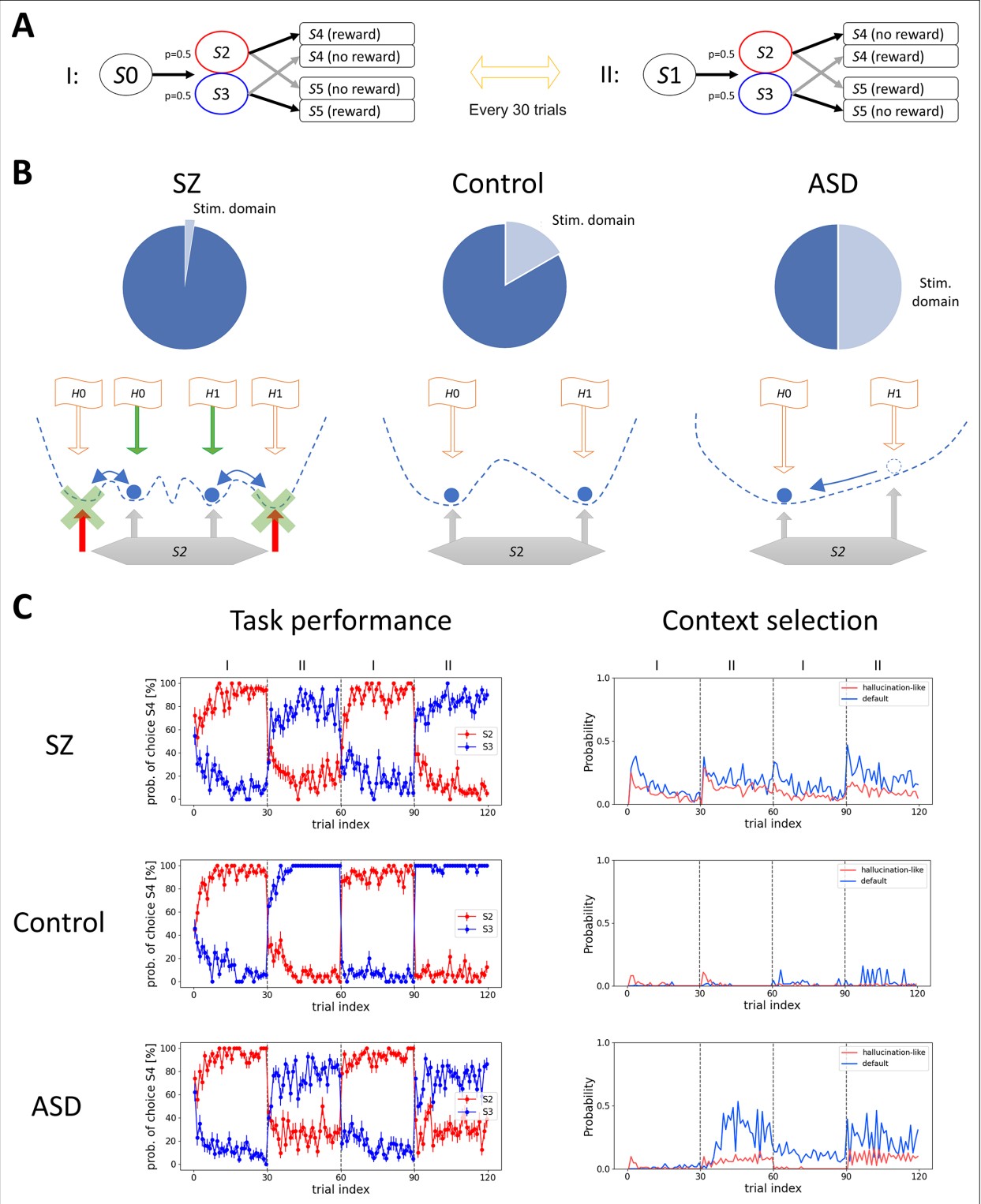

**Figure 5.** Model prediction about the relationship between sensory processing and flexible behavior. (**A**) Task diagram. The structure is the same as *Figure 4*, but the probability of *S2* and *S3* is equal. (**B**) (Top) We tested three stimulus neuron ratios: 2.5% for schizophrenia (SZ), 16.7% for control, and 50% for autism spectrum disorder (ASD). (Bottom) Schematics of how Context selector changes by the manipulation of neuron ratios in this task. Blue dotted lines indicate the energy landscape, and blue circles indicate the attractor dynamics. Red arrows indicate the wrong stimulus prediction (hallucination-like effects) which triggers sensory prediction error (SPE)-driven remapping (green cross marks and arrows), and orange lines indicate the input from the hippocampus to *X* (*H*0 and *H*1 indicate hippocampal segments in *S*0 and *S*1, respectively). (**C**) (Left) The probability of choosing *S4* at *S2* and *S3* is plotted in red and blue, respectively. The error bars indicate the standard error of the mean (*N*=40). The SZ model fails to show one-shot

*Figure 5 continued on next page*

*Figure 5 continued*

switch for the second experience in environments I and II, while the ASD model shows an impaired task performance mainly in environment II. (Right) The result of context selection (see *Figure 1—figure supplement 1*). The probability of wrong stimulus reconstruction (hallucination-like) is plotted in red, and the probability of default context usage due to failures in context reconstruction (see Materials and methods) is plotted in blue.

The online version of this article includes the following figure supplement(s) for figure 5:

**Figure supplement 1.** Reward-dependent plasticity when sensory and contextual encoding neurons coexist in the hippocampus.

error-based remapping, SPE-driven remapping, and RPE-facilitated remapping as a key for generating context-dependent sequential activity change in the hippocampus that enables flexible behavior. This mechanism is biologically plausible, as it is observed in the hippocampus (*Bostock et al., 1991*) and in some cortical regions (*Castegnetti et al., 2021*). Our model could simulate a variety of context-dependent sequential representations in the hippocampus such as splitter cells (*Wood et al., 2000*), lap cells (*Sun et al., 2020*), probabilistic model selection (*Ekman et al., 2022*), and contextual inference (*Julian and Doeller, 2021*), without task-dependent parameter tuning. Furthermore, our model predicted a mechanistic explanation for the co-occurrence of deficits in sensory processing and flexible behavior. This result is supported by the clinical reports that psychosis can change the attractor dynamics in the hippocampus (*Rolls, 2021*), and treatments for sensory processing helped restore flexible behavior in some psychoses (*Andelin et al., 2021*; *Javitt and Freedman, 2015*; *Pfeiffer et al., 2011*; *Reed et al., 2020*). To the best of our knowledge, this is the first model that uses associative memory for describing the formation and switching of context-dependent hippocampal activity through remapping and its contribution to flexible behavior.

Our model is a functionally modular account of the cortical regions and hippocampus, enabling it to capture experimental findings across species. While hippocampal activity in rodents has been extensively characterized in terms of spatial coding, human hippocampal representations are more often non-spatial and episodic-like (*Bellmund et al., 2018*; *Eichenbaum, 2017*). For episodic memory to support flexible behavior, it would be beneficial to retrieve each episode in a context-dependent manner. The episodic contents may vary across species and individuals, yet the fundamental computations—estimating the current context from external stimuli and their history, and flexibly updating this estimate via prediction errors—are likely conserved. Holding context information until the contextual prediction error is detected is analogous to the belief state in model-based reinforcement learning, which is known to improve performance under partially observable conditions (partially observable Markov decision processes [POMDPs]) (*Kaelbling et al., 1998*). Our model provides a simple algorithmic implementation of this principle.

Although remapping is a widely known phenomenon, its mechanism remains under debate. We used the Amari-Hopfield network as Context selector to distinguish multiple contextual states that share the same external stimuli and to reconstruct them via attractor dynamics from partial observations. We propose two advantages of this associative memory model. First, it can represent different contexts under the same external stimuli depending on the feedback from $H$ to implement rapid behavioral switching without requiring synaptic changes. The second advantage is its ability to infer a contextual state using the associative memory mechanism. This property might occasionally yield a non-trivial contextual state based on past experiences. Expanding upon our model with more sophisticated associative memory search mechanisms could enable creative behavior.

We speculate that Context selector is implemented across multiple brain regions with varying degrees of resolution, including a part of the EC and PFC. First, the lateral EC provides item-specific and sensory context information (*Deshmukh and Knierim, 2011*; *Hargreaves et al., 2005*), whereas the medial EC supplies history information and state signals (*Hafting et al., 2005*; *Heys and Dombeck, 2018*). Because these inputs jointly shape hippocampal attractor dynamics, the EC is well positioned to determine which subjective context is selected. Second, PFC has been reported to retain context-dependent attractors, which reflect working memory (*D'Ardenne et al., 2012*), attention (*Siegel et al., 2015*), and confidence (*Wynn and Nyhus, 2022*), and to send inputs to the hippocampus. In addition, the PFC computes prediction errors that might trigger remapping. Specifically, reward-related prediction errors are computed in the orbitofrontal cortex (OFC) (*Garvert et al., 2023*; *Stalnaker et al., 2014*), anterior cingulate cortex (ACC) (*Seo and Lee, 2007*), and ventromedial PFC (*Rehbein et al., 2023*), whereas stimulus-related prediction errors are calculated in the ACC (*Ide et al., 2013*) and dorsolateral PFC (*Masina et al., 2018*; *Zmigrod et al., 2014*). These neural circuits

likely coordinate to estimate the current context and select the appropriate representation in the hippocampus via remapping. Our modeling of Context selector captures this core functionality in a simplified manner. Incorporating more elaborate features, such as multiple hierarchies (*Rao, 2024*), in future studies might help explain a broader range of experimental results.

Our model posits that the Sequence composer corresponds to computations within the hippocampus. As a biologically plausible projection, we consider the CA3-CA1 circuit, where contextual inputs from regions such as the PFC and EC provide the current contextual state to CA3, enabling the recurrent CA3-CA1 architecture to generate predictions of the next contextual state without errors in action. Consistent with this idea, the temporal lag in CA3→CA1 transmission suggests a functional gradient in which CA3 represents present-oriented information while CA1 carries more future-oriented predictions (*Chen et al., 2024*), and neurons in both CA3 and CA1 exhibit action-driven remapping and encode action-planning signals (*Green et al., 2022*). Our framework, therefore, predicts that changes in CA3→CA1 population activity precede behavioral switching in context-dependent alternation in *Figure 2* or multi-lap tasks in *Figure 3*, and perturbation of this input will degrade the behavioral performance.

Beyond the function of individual components described above, our framework also yields several predictions about how these regions interact to support flexible behavior. We propose three experiments. First, our model posits that an error about the context triggers remapping. The OFC is known to be active when reward-related prediction error occurs (*Banerjee et al., 2020*), and hippocampal remapping is suggested to be induced by the EC, especially its lateral part (*Latuske et al., 2017*). Because a direct projection exists from the OFC to the lateral EC (*Kondo and Witter, 2014*), this input might critically influence hippocampal remapping. Second, our model suggests that the prediction error about the environment would induce a shift from place-cell encoding to lap-cell encoding in the hippocampus (*Figure 3*). Third, our model proposes two types of prediction error: one is the conventional prediction error that updates the synaptic weights within the context, and the other is the prediction error about the context that triggers remapping in *X* and *H*. How these two different prediction errors are represented in neural circuits will deepen our understanding of the neural basis of flexible behavior.

Our model also provides an algorithmic-level account of psychiatric symptoms by changing the relative weighting of sensory-encoding versus context-coding neurons. This implementation is analogous to Bayesian theories linking priors to psychiatric symptoms. In SZ, hallucinations and delusions have been modeled as arising from overly strong top-down priors (*Powers et al., 2016*) or circular inference, which leads to erroneous belief formation (*Jardri et al., 2017*; *Jardri and Denève, 2013*). In our model, we used an underrepresented stimulus domain to increase the relative influence of internally generated context representation in context selection. Crucially, this implementation does not simply strengthen priors but induces excessive generation and competition of contextual states, leading to frequent yet non-reproducible remapping of hippocampal contextual activity and a failure of learning to converge despite repeated experience. In ASD, it has been argued that abnormally high sensory precision reduces the updating of expectations (*Karvelis et al., 2018*) or leads to sensory-dominant perception, which has been interpreted as weak priors (*Angeletos Chrysaitis and Seriès, 2023*; *Lawson et al., 2014*; *Pellicano and Burr, 2012*). In our framework, we used an overrepresented stimulus domain to increase the relative influence of external stimulus representations in context selection. Importantly, our model captures not only sensory-dominant processing emphasized in previous studies, but also a distinctive impairment in flexibly utilizing newly introduced contexts, reflecting a failure of context reconstruction and resulting in persistent inflexible behavior. Thus, our conjunctive modeling of sensory and context processing complements Bayesian accounts of psychiatric symptoms and provides a mechanistic explanation for the role of sensory processing in maladaptive, inflexible behavior.

Our model also has limitations. First, there are context-dependent tasks that our model cannot solve. Although our model learns to separate contextual states, it does not combine them; consequently, we did not consider simulating the environment in which the number of hidden states decreases over time. Greater flexibility might be achieved by integrating both sensory and contextual information within certain neurons (e.g. *Figure 5—figure supplement 1*). Second, the resolution at which our model should distinguish different contextual states, including the stimulus resolution and time resolution, is hand-tuned in this work. While we used an abstract, grid-like state space with discrete time,

an important direction for future work is to model its activity at finer-grained neural timescales, such as theta cycles (*Foster and Wilson, 2007*; *Wikenheiser and Redish, 2015*). In realistic, continuously changing environments, such resolutions should be adjusted autonomously. Introducing continuous and hierarchical representations with multiple levels of spatial and temporal resolution would facilitate such adjustments, potentially through mechanisms such as modern Hopfield networks (*Krotov and Hopfield, 2020*) or synfire-chain-based hippocampal sequence generation (*Abeles, 1982*; *Diesmann et al., 1999*; *Shimizu and Toyoizumi, 2025*; *Toyoizumi, 2012*), but this is beyond the focus of the current study. Third, our model assumed that only the hippocampus projects to the midbrain for reward prediction of sequential plans. However, there are projections from other brain regions, including the cortex, to the midbrain that are also involved in reward prediction (*Jo and Mizumori, 2016*). How these additional projections influence model-based behavior, especially in the case of hippocampal lesions, remains beyond the scope of this work. Finally, explicitly modeling the input from grid cells that encode geometric task structure (*Krupic et al., 2015*) might enable more sophisticated planning (e.g. discovering the shortest path).

# Materials and methods

## Simulation environment

We conducted all simulations and post hoc analysis using a custom-made Python code.

## Model description

### Overview

Below, we introduce a model that describes the acquisition of model-based reasoning. Our model consists of two components: Context selector ($X$) and Sequence composer (hippocampus, $H$). For simplicity, the environment is defined in discrete time, and agents move through environmental states characterized by distinct external stimuli. The model's computational dynamics are fundamentally synchronized with the environmental (behavioral) time step, and at each time step, the agents transition to the next environmental state. Upon a state transition, the agents first perform contextual state estimation by Context selector and activate a corresponding hippocampal neuron. Then, this hippocampal neuron initiates sequential activity based on hippocampal synaptic connectivity. Each hippocampal sequence represents a planned course of action and is used to predict a series of external stimuli. The agents follow the plan unless SPE-driven remapping (see 'SPE-driven remapping' section) or RPE-facilitated remapping (see 'RPE-facilitated remapping' section) occurs. The hippocampal sequence from which actions are generated is updated upon a reward. After the action execution, the agents repeat the process by selecting the current contextual state. As the agents become familiar with the environment, hippocampal sequences that enable future predictions become longer, and contextual state estimation by Context selector becomes less frequent. The algorithmic flowchart of our model is described in *Figure 1—figure supplement 1*.

### Context selector (X)

We model Context selector as an Amari-Hopfield network (*Amari, 1972*; *Hopfield, 1982*) of $N$=1200 binary neurons, whose activity is described by vector $X$. We employ the Amari-Hopfield model because it allows multiple contexts to be stably maintained in response to stimuli and can be trained via Hebbian plasticity. We assume that similar computations are carried out in the PFC and EC circuits in the brain.

$X$ consists of two domains: stimulus domain $X^{stim}$ and context domain $X^{cont}$. The neuron ratio in the stimulus domain over the whole neurons $dim\left(X^{stim}\right)$/$N$ is 16.7% (200 neurons) for the control condition, 2.5% (30 neurons) for the SZ condition, and 50% (600 neurons) for the ASD condition. Note that $dim$ describes the dimensions of a vector.

When the agents visit each environmental state for the first time, the $X$'s activity is set to

$$X=\begin{pmatrix} X^{stim} \\ X^{cont} \end{pmatrix} = \begin{pmatrix} \xi^{stim} \\ f\left(\xi^{stim}\right) \end{pmatrix} \qquad (1)$$

where the converter function $f\left(\xi^{stim}\right)=binary\left(A\xi^{stim}gta\right)$ returns a binary vector computed from $dim\left(X^{cont}\right)$ by $dim\left(X^{stim}\right)$ default matrix $A$ with independently and identically distributed unit Gaussian entries and scalar threshold $a$ chosen so that $f\left(\xi^{stim}\right)$ consists of half 1 and half 0 elements. This contextual state is set as a default context, ensuring that the $X$ module assigns a unique contextual state to each environmental state. Biologically, one possible interpretation is that this default context corresponds to modality-specific innate representations in prefrontal regions (**Manita et al., 2015**).

From the second visit of each environmental state after completing actions according to a hippo-campal sequence, the contextual state is determined by associative memory dynamics of the Amari-Hopfield network. We adopt two ways of initialization: history-based and landmark-based (see **Figure 1—figure supplement 1**). While the history-based initialization was introduced to select the contextual state based on the history input from $H$, the landmark-based initialization was introduced to terminate the episodic sequence that would otherwise continue indefinitely. Biologically, the landmark-based initialization corresponds to the operation of anchoring a contextual state to salient environmental landmarks—such as an animal's nest—that serve as clear reference points. Formally, we use the history-based initialization when the input from $H$ to $X$ predicts the next contextual state, i.e., $W^{XH}H$ is not all zero, where $W^{XH}$ represents the synaptic weights from $H$ to $X$. We use the landmark-based initialization when the input from $H$ to $X$ does not provide any predictive input, i.e., $W^{XH}H$ is all zero, but they are at a landmark (we defined it as the initial environmental state of each task). When the inputs from $H$ to $X$ do not predict the next contextual state and agents are not at a landmark (history mismatch), which typically happens after remapping, a new contextual state is generated and stored in the Amari-Hopfield network (see 'SPE-driven remapping' and 'RPE-facilitated remapping' sections). Biologically, these distinctions could naturally arise from the interplay of the strength of history-dependent inputs, sensory saliency, and the depth of contextual attractors, which would be dynamically integrated in the PFC and EC circuits.

The history-based initialization starts from the initial state of the Amari-Hopfield network

$$X_{init}=binary\left(W^{XH}H>0\right) \tag{2}$$

where $binary$ represents the indicator function that takes 1 if the argument is true and 0 otherwise. The landmark-based initialization starts from the initial state of the Amari-Hopfield network

$$X_{init}=\begin{pmatrix} \xi^{stim} \\ random \end{pmatrix} \tag{3}$$

where $random$ indicates a random binary vector consisting of half 0 and half 1 elements.

After history-based or landmark-based initialization, $X$ iteratively updates its contextual state at the beginning of each time step according to the associative memory dynamics:

$$X \leftarrow binary\left(W^{XX}\left(X-X^0\right)-\theta\right) \tag{4}$$

where $\theta=0.5$, $X^0=0.5$, and $dim\left(X\right)$ by $dim\left(X\right)$ matrix $W^{XX}$ represents synaptic weights of Context selector (see 'Synaptic weight update' section for how $W^{XX}$ changes). These dynamics end up either as a successful or failed recall. A recall is defined as successful if $X$ converges within 50 iterations, and its stimulus domain $X^{stim}$ becomes identical to $\xi^{stim}$. If $X$ fails to converge within 50 iterations, the contextual state is set to the default contextual state defined in **Equation 1**. This default implemen-tation is analogous to psychological inertia, particularly under uncertainty (**Ip and Nei, 2025**; **Sautua, 2017**), which has been reported to be more pronounced in ASD patients (**Joyce et al., 2017**). If $X$ converges within 50 iterations but the stimulus domain $X^{stim}$ of the converged $X$ is different from $\xi^{stim}$ (hallucination-like effects), agents consider that they are in a new context, and SPE-driven remapping occurs (see **Figure 1S**). Reuse of the default contextual state and the hallucination-like effects become critical for explaining ASD and SZ phenotypes, respectively. As one possible biological implementa-tion, we consider context selection in $X$ as a brain-wide evoked potential during which bottom-up information may be integrated with top-down signals to select the current context (**Mohanty et al.,**

2025). In this case, it takes several hundred milliseconds for the contextual states in $X$ to settle (*Massimini et al., 2005*).

After $X$ is set, the agents randomly generate a hippocampal sequence reflecting it (see 'Sequence composer (hippocampus, $H$)' section). Then, the agents evaluate this sequence that encodes a course of actions and act according to it (see 'Action flow' section).

## Sequence composer (hippocampus, $H$)

We model Sequence composer (hippocampus) with $N$=300 binary RNN. The hippocampus produces sequential activity probabilistically based on the contextual state computed above. Starting from the seed hippocampal neuron directly activated by the contextual state, the next hippocampal neuron is iteratively activated with a probability proportional to the synaptic weights from the previously activated hippocampal neuron. Therefore, the same contextual state could generate diverse sequences. This randomness in the sequence generation facilitates the exploration behavior of the agents, which is important for reinforcement learning, but also adds noise to the input from Sequence composer to Context selector in the history-based computation.

Hippocampal neurons initially receive input vector $W^{HX}X_k$, where $W^{HX}$ is the synaptic weight matrix from $X$ to $H$, and $X_k$ is the contextual state at time step $k$. Only the neuron that receives the strongest input is activated, whose index is described as

$$\widetilde{H}_k^{(S)} = argmax\left(W^{HX}X_k\right) \tag{5}$$

(see 'Synaptic weight update' section for how $W^{HX}$ changes), where the tilde mark indicates a neuron index.

Our model has two types of hippocampal neurons: state-coding and transition-coding types. The indices of neurons belonging to these types are denoted as $\widetilde{H}^{(S)}$ and $\widetilde{H}^{(T)}$, respectively. The state-coding neurons receive input from $X$ and represent the current contextual state, while the transition-coding neurons send output to $X$ and predict the next contextual state after an action, i.e., $T(X_{k+1}|X_k, a_{k,k+1})$. One possible biological grounding for this functional separation is that the EC provides contextual inputs to CA3, and CA3 and CA1 generate predictions of the next state through its recurrent architecture (*Chen et al., 2024*). Also, neurons in CA3 and CA1 are reported to show action-driven remapping to be involved in action planning (*Green et al., 2022*). When the agents experience a contextual state $X_k$ for the first time, $\widetilde{H}_k^{(T)}$ is randomly chosen and the synaptic weight from $\widetilde{H}_k^{(S)}$ to $\widetilde{H}_k^{(T)}$ is set to 1. From the second experience of the contextual state $X_k$, the corresponding hippocampal neuron $\widetilde{H}_k^{(S)}$ initiates a sequence $H=$ of hippocampal activity with a non-negative integer $\tau$, where the next neuron is recursively chosen with a probability vector proportional to

$$\frac{\left[W^{HH}\right]_{\cdot\widetilde{H}_k} - w_0}{1 - w_0} binary\left(\left[W^{HH}\right]_{\cdot\widetilde{H}_k} - w_0 > 0.01\right) \tag{6}$$

where $\left[W^{HH}\right]_{\cdot\widetilde{H}_k}$ describes a vector of intra-hippocampal synaptic weights from neuron $\widetilde{H}_k$ and $w_0$=0.3 is the effective threshold. The sequential activity can stop at a transition-coding neuron $\widetilde{H}_{k+\tau}^{(T)}$ according to two conditions: when all the synaptic weights $\left[W^{HH}\right]_{\cdot\widetilde{H}_{k+\tau}^{(T)}}$ are equal to or below 0.01 or when the reward value function of the lastly activated transition-coding neuron $\widetilde{H}_{k+\tau}^{(T)}=\widetilde{H}_{-1}$ becomes positive (see 'Reward prediction' section).

The synaptic connection from a state-coding neuron to a transition-coding neuron is formed in a reward-independent manner as described above, whereas the connection from a transition-coding neuron to a state-coding neuron is established in a reward-dependent manner (see 'Synaptic weight update' section). Consequently, when animals receive few rewards during the initial exploration phase, minimal sequences with $\tau$=0 are constructed. As animals discover rewarding behaviors, these minimal sequences join, and eventually, agents anticipate the rewarding transition ahead.

When the number of contextual states increases particularly in the SZ condition, representational overlap arises between hippocampal state-coding and transition-coding neurons. This overlap makes

the prediction of the next contextual state by the transition-coding neurons unreliable. The degraded prediction from $H$, in turn, corrupts the initial condition for context selection in $X$ (*Equation 3*), leading to hallucination-like behavior.

## Reward prediction

Each hippocampal sequence $H$ is associated with rewards, perhaps via the operation of the midbrain. Reward value function $V_{\widetilde{H}_{-1}}$, which depends on the lastly activated transition-coding hippocampal neuron $\widetilde{H}_{-1}$ of the sequence, is updated every time the agents receive reward $R > 0$ according to

$$V_{\widetilde{H}_{-1}} \leftarrow V_{\widetilde{H}_{-1}} + \alpha \left( R - V_{\widetilde{H}_{-1}} \right) \tag{7}$$

with learning rate $\alpha = 0.15$. The sequence value $SV_{\widetilde{H}_{-1}}$ associated with $\widetilde{H}_{-1}$ mirrors $V_{\widetilde{H}_{-1}}$ except when it is suppressed by this neuron's *no-good* indicator $NG_{\widetilde{H}_{-1}}$ (cross marks in *Figure 1C*), namely,

$$SV_{\widetilde{H}_{-1}} = V_{\widetilde{H}_{-1}} - \left( V_{\widetilde{H}_{-1}} + NG_{\widetilde{H}_{-1}} \right) \cdot binary \left( NG_{\widetilde{H}_{-1}} \geq \theta_{NG} \right) \tag{8}$$

where suppression threshold $\theta_{NG}$ is set to 0.7. *No-good* indicator is introduced to transiently suppress previously established sequences that have not been recently rewarded, without devaluing them. This *no-good* indicator facilitates RPE-facilitated remapping (see 'RPE-facilitated remapping' section) that leads to exploration of different contextual states in $X$ and sequences in $H$. The *no-good* indicator is inspired by recent findings in the ventral hippocampus, where dopamine D2-expressing neurons of the ventral subiculum selectively promote exploration under anxiogenic contexts (*Godino et al., 2025*). When the *no-good* indicator is active, i.e., $NG_{\widetilde{H}_{-1}} \geq \theta_{NG}$, the sequence value becomes transiently negative. Note that we set $\theta_{NG}$ as 0.7 to make the agents sufficiently sensitive to abrupt environmental changes and enable exploring some candidate contexts after RPE-facilitated remapping.

These neurons' *no-good* indicators change when a reward is presented. The *no-good* indicator of the lastly activated hippocampal neuron $\widetilde{H}_{-1}$ instantaneously drops to 0 when the reward is greater than the reward value function, i.e., $R > V_{\widetilde{H}_{-1}}$ but instantaneously increases by 1 otherwise. In addition, the *no-good* indicators of all hippocampal neurons gradually decay according to

$$NG \leftarrow \gamma NG \tag{9}$$

with multiplication factor $\gamma = 0.7$ when the reward is less than the reward value function, i.e., $R < V_{\widetilde{H}_{-1}}$.

## Action flow

After completing each environmental state transition according to a planning sequence without remapping (see 'SPE-driven remapping' and 'RPE-facilitated remapping' sections), the agents estimate a contextual state (context selection, *Figure 1—figure supplement 1*) and, based on it, generate a hippocampal sequence $H$ (sequence composition, *Figure 1—figure supplement 1*). Below, we describe how the agents select one hippocampal sequence. The last hippocampal neuron $\widetilde{H}_{-1}$ of the sequence $H$ informs its sequence value $SV_{\widetilde{H}_{-1}}$ (see 'Reward prediction' section). When $SV_{\widetilde{H}_{-1}}$ is positive, the agents select this sequence. Otherwise, the agents reject this hippocampal sequence and compose another hippocampal sequence (using a different random seed for the landmark-based initialization) for up to nine attempts (sequence selection loop, *Figure 1—figure supplement 1*). If none of the nine sequences have positive $SV_{\widetilde{H}_{-1}}$, one is selected randomly, excluding that with the lowest sequence value. We use this sequence selection to provide a balance between exploration and exploitation of sequence selection and serve as a good compromise for visualization. Once a sequence is selected, agents start to execute sequence-based action loop (see *Figure 1—figure supplement 1*) unless RPE-facilitated remapping (see 'RPE-facilitated remapping' section). The transition-coding hippocampal neurons $\left[ \widetilde{H}_k^{(T)}, \cdots, \widetilde{H}_{k+\tau}^{(T)} \right]$ in the sequence specify the transition of environmental states (action), i.e., the stimulus domain of the input from $H$ to $X$ ($binary \left( w^{XH} H^{(T)} > 0 \right)$) represents the prediction of the next

environmental states, where $W^{XH}$ is the synaptic weights from $H$ to $X$ and $H^{(T)}$ is the hippocampal state when each transition-coding neuron is active. The sequence with $\tau$ transition-coding neurons provides an action plan in the next $\tau$ steps, unless SPE-driven remapping happens (see 'SPE-driven remapping' section). This is inspired by preplay or planning by hippocampal sequences (**Dragoi and Tonegawa, 2011**). After completing the final action specified by the sequence, the agents repeat the whole procedure, starting from the contextual state estimation.

## SPE-driven remapping

SPE-driven remapping can occur while the agents execute a course of actions following a hippocampal sequence. We refer to SPE-driven remapping as the shift of $X$'s activity to another contextual state or generate a new one under the same external stimuli. Upon the course of actions following hippocampal sequence $H=$, the prediction of the next external stimuli, i.e., the stimulus domain of $binary\left(W^{XH}H^{(T)} > 0\right)$, may differ from the actual one, $\xi^{stim}$. When this happens at a sequence location $k+\tau'$ ($1 \leq \tau' \leq \tau$), SPE-driven remapping (**Figure 1D**) occurs, and the synaptic weights in $X$ and $H$ are modified. If the event occurs during $1 \leq \tau' < \tau$, steps 1–3 are applied. In contrast, if the event occurs at $\tau'=\tau$, only step 3 is applied.

1. The hippocampal sequence is interrupted between the transition $\overset{\sim}{H}^{(T)}_{k+\tau'-1} \to \overset{\sim}{H}^{(S)}_{k+\tau'}$, and the corresponding synaptic weight is weakened (see 'Synaptic weight update' section).

2. If the transition-coding neuron $\overset{\sim}{H}^{(T)}_{k+\tau'-1}$ projects to state-coding neurons other than $\overset{\sim}{H}^{(S)}_{k+\tau'}$, these state-coding neurons' predictions about external stimuli are examined. If there exists one that predicts the actual external stimuli with an error less than the remapping threshold of $\theta_{remap}=5\,bit$, this neuron is activated, and the contextual state $X$ is set based on its input (**Equation 2**, 'Context selector ($X$)' section). Otherwise, step 3 is applied. Note that we set the remapping threshold $\theta_{remap}=5\,bit$ to allow for small mis-convergence during recall in the Amari-Hopfield model.

3. A new contextual state is set as $X=\left(\xi^{stim}, random\right)^{\top}$ with the synaptic weights $W^{XX}$ updated (see 'Synaptic weight update' section). A hippocampal neuron is activated based on the new contextual state in $X$ following **Equation 5**, and the synaptic weight is strengthened between the interrupted transition-coding hippocampal neuron $\overset{\sim}{H}^{(T)}_{k}$ and the newly activated state-coding hippocampal neuron (see 'Synaptic weight update' section).

When a new hippocampal neuron is recruited in step 3, the history mismatch occurs in the following environmental state because this hippocampal neuron does not predict the upcoming external stimulus. Therefore, once SPE-driven remapping is triggered, the contextual states in $X$, as well as the activated neurons in $H$, are repeatedly updated in the following environmental states until the agents encounter a landmark (i.e. starting point) and reset the episode.

## RPE-facilitated remapping

To gain information on the environment, the agents perform exploration followed by RPE-facilitated remapping. We refer to exploration as a random action not specified by the selected sequence. Exploration can occur with probability $p_{expl}$ whenever the agents enter an environmental state with the number of transition candidates greater than the number of transition-coding hippocampal neurons initiating from the corresponding state-coding hippocampal neuron. The exploration probability is generally $p_{expl}=0.3$ but increases to certainty ($p_{expl}=1$) if the agents are taking actions following a sequence with a negative sequence value, which happens when its *no-good* indicator is active, i.e., $NG_{\overset{\sim}{H}_{-1}} \geq \theta_{NG}$. In case of this exploration, one of the unconnected transition-coding hippocampal neurons is randomly activated (RPE-facilitated remapping), and the agents take a random transition. At the following environmental state, $X$ is set to be a random contextual state $X=\left(\xi^{stim}, random\right)^{T}$, and synaptic weights of $H$ and $X$ are updated (see 'Synaptic weight update' section).

Same as SPE-driven remapping, once RPE-facilitated remapping is triggered, the history mismatch occurs and the contextual states in $X$, as well as the activated neurons in $H$, are repeatedly updated in the following environmental states until the agents encounter a landmark (i.e. starting point) and reset the episode.

## Synaptic weight update

We used a Hebbian learning rule to update the synaptic weight matrix $W^{XX}$ only for the first time contextual state $X$ is settled:

$$W^{XX} \leftarrow W^{XX} + \left( X - X^0 \right) \left( X - X^0 \right)^{\top} \tag{10}$$

We also used a basic Hebbian learning rule for updating synaptic weights between $X$ and $H$. Again, only for the first time a hippocampal neuron is activated according to *Equation 5* in response to contextual state $X_k$, synaptic weights are updated as

$$W^{HX} \leftarrow W^{HX} + \eta H^{(S)} \left( X_k - X^1 \right)^{\top} \tag{11}$$

$$W^{XH} \leftarrow W^{XH} + \eta \left( X_k - X^1 \right) \left( H^{(S)} \right)^{\top} \tag{12}$$

$$W^{XH} \leftarrow W^{XH} + \eta \left( X_k - X^1 \right) \left( H^{(T)}_{-1} \right)^{\top} \tag{13}$$

where $H^{(S)}$ and $H^{(T)}$ are the state-coding and transition-coding hippocampal activity vectors, respectively, whose elements take 1 for the activated neuron of the corresponding type and 0 for the others. Note that the initial synaptic weights of $W^{HX}$ and $W^{XH}$ are all 0. Similarly, $H^{(T)}_{-1}$ is the transition-coding hippocampal activity vector of the previous hippocampal sequence, where the element corresponding to the last transition-coding neuron takes 1, and others take 0. Learning rate $\eta = (N+1)/2$ and offset $X^1 = N/(N+1)$ are chosen to achieve good association dynamics in Context selector. These synaptic weights change within the bound $W^{XH}, W^{HX} \leq 1/2$.

We used different learning rules for the intra-hippocampal synaptic weights depending on within-episodic and between-episodic segments. The initial synaptic weights are all $w_0$, and these weights change within the bound $0 \leq W^{HH} \leq 1$. Within-episodic connections, i.e., state-coding to transition-coding synapses, are constantly updated in a reward-independent manner when $\widetilde{H}_k^{(S)}$ and $\widetilde{H}_k^{(T)}$ are activated as

$$W^{HH} \leftarrow W^{HH} + (1 - w_0) H_k^{(T)} \left( H_k^{(S)} \right)^{\top} - 0.5\alpha H_k^{(T)} \left( 1 - H_k^{(S)} \right)^{\top} binary \left( \left[ W^{HH} \right]_{\widetilde{H}_k^{(T)} \widetilde{H}_k^{(S)}} \leq w_0 \right) \tag{14}$$

The second term describes Hebbian potentiation, and the third term describes hetero-synaptic depression between non-active presynaptic neurons and the active postsynaptic neuron. Note that we assume hetero-synaptic depression only upon the initial establishment of the synaptic connection between the two hippocampal neurons. This modeling is inspired by behavioral timescale plasticity in the hippocampus (*Bittner et al., 2017*), in which synaptic potentiation occurs for events that are close in time regardless of reward, and such plasticity is believed to support the formation of place cells, etc. Between-episodic connections, i.e., transition-coding to state-coding synapses, are constantly updated in a reward-dependent manner when the agents receive a reward ($R > 0$) and $\widetilde{H}_k^{(T)}$ and $\widetilde{H}_{k+1}^{(S)}$ are involved in $H$ according to

$$W^{HH} \leftarrow W^{HH} + \alpha \left( R - \left[ W^{HH} \right]_{\widetilde{H}_{k+1}^{(S)} \widetilde{H}_k^{(T)}} - w_0 \right) H_{k+1}^{(S)} \left( H_k^{(T)} \right)^{\top}$$
$$- 0.5\alpha H_{k+1}^{(S)} \left( 1 - H_k^{(T)} \right)^{\top} binary \left( \left[ W^{HH} \right]_{\widetilde{H}_{k+1}^{(S)} \widetilde{H}_k^{(T)}} \leq w_0 \right) \tag{15}$$

The second term describes Hebbian potentiation that modifies the weight $\left[ W^{HH} \right]_{\widetilde{H}_{k+1}^{(S)} \widetilde{H}_k^{(T)}}$ toward $R - w_0$, and the third term describes hetero-synaptic depression. This is supported by the finding that dopaminergic neuromodulation gates LTP, enabling preferential consolidation of reward-associated experiences (*Lisman and Grace, 2005*; *Takeuchi et al., 2016*).

In addition, if SPE-driven remapping happens at the sequence location between $\widetilde{H}_k^{(T)}$ and $\widetilde{H}_{k+1}^{(S)}$, the synaptic weight from $\widetilde{H}_k^{(T)}$ to $\widetilde{H}_{k+1}^{(S)}$ is weakened by $-\alpha \left[ W^{HH} \right]_{\widetilde{H}_{k+1}^{(S)} \widetilde{H}_k^{(T)}}$, while that from $\widetilde{H}_k^{(T)}$ to the activated state-coding hippocampal neurons $\widetilde{H}_{k+1}^{(S)}$, is strengthened by

$$\alpha \left( 0.65 - \left[ W^{HH} \right]_{\widetilde{H}_{k+1}^{(S)} \widetilde{H}_k^{(T)}} \right) binary \left( \left[ W^{HH} \right]_{\widetilde{H}_{k+1}^{(S)} \widetilde{H}_k^{(T)}} < 0.65 \right).$$

Considering the memory capacity of the Amari-Hopfield network with correlated patterns, the number of memorizable contextual states sharing the same external stimulus is below 8. If this condition is violated, to prevent overloading the Amari-Hopfield network, the contextual state $X$ that has never produced hippocampal sequences with a sequence value more than 0.7 induces a forgetting process as

$$W^{XX} \leftarrow W^{XX} - \left( X - X^0 \right) \left( X - X^0 \right)^{\top} \tag{16}$$

This process represents forgetting of reward-unrelated episodic memory.

## Formal descriptions of each task setting

All tasks used in this study were formulated as POMDPs, defined as

$$M = \langle S, A, T, R, C \rangle$$

Below, we describe the model components for each task.

### Alternation task (*Figure 2*)

The alternation task (*Figure 2*) can be described as follows.

- **State space** $S = \{S_1, S_2, S_3, S_4, S_5\}$, where $S_1$ is the starting point, and $S_4$ and $S_5$ are the reward delivery points.
- **Action space** $A = \{a_{12}, a_{24}, a_{25}, a_{32}, a_{43}, a_{51}\}$, where each action determines a state transition.
- **Transition function** $T\left(S_j | S_i, a_{ij}\right) = 1$, specifying the probability of reaching state $S_j$ given current state $S_i$ and action $a_{ij}$. In this task, transitions are deterministic given the correct context, but ambiguous without context.
- **Reward function** $R_t(s, c_t) = \begin{cases} 1 & \text{if } (s = S_4,\ c_t = 1) \\ 1 & \text{if } (s = S_5,\ c_t = 2) \\ 0 & \text{otherwise} \end{cases}$, where $t$ indicates the trial index, and $c_t$ indicates the hidden state at trial $t$.
- **Hidden state** $c_t = \begin{cases} 1 & \text{if } c_{t-1} = 2 \text{ and } R_{t-1} = 1 \\ 2 & \text{if } c_{t-1} = 1 \text{ and } R_{t-1} = 1 \end{cases}$, where hidden variable switches depending on the previous reward under initial condition of $c_1 = 1$.

### 2-Lap task (*Figure 3*)

2-Lap task (*Figure 3*) can be described as follows.

- **State space** $S = \{S_1, S_2, S_3, S_4\}$, where $S_1$ is the starting point, and $S_4$ is the reward delivery point.
- **Action space** $A = \{a_{12}, a_{23}, a_{24}, a_{32}\}$, where each action determines a state transition.
- **Transition function** $T\left(S_j | S_i, a_{ij}\right) = 1$, specifying the probability of reaching state $S_j$ given current state $S_i$ and action $a_{ij}$. In this task, transitions are deterministic given the correct context, but ambiguous without context.

- **Reward function** $R_t(s, c_t) = \begin{cases} 1 & \text{if } (s = S_4, N_t(S_3) = 0, c_t = 1) \\ 1 & \text{if } (s = S_4, N_t(S_3) = 1, c_t = 2) \\ 1 & \text{if } (s = S_4, N_t(S_3) \geq 2, c_t = 3) \\ 0 & \text{otherwise} \end{cases}$, where $t$ indicates the trial index,

$N_t(S3)$ indicates the number of visiting $S3$ at trial $t$, and $c_t$ indicates the hidden state at trial $t$.

- **Hidden state** $c_t = \begin{cases} 1 & \text{if } t \leq 20 \\ 2 & \text{if } 20 < t \leq 40 \\ 3 & \text{if } t > 40 \end{cases}$, where hidden variable switches depending on the trial

index.

Note that in **Figure 3G and H**, we used the following reward function and hidden state.

- **Reward function** $R_t(s, c_t) = \begin{cases} 1 & \text{if } (s = S_4, N_t(S_3) = 1, c_t = 2) \\ 1 & \text{if } (s = S_4, N_t(S_3) \geq 2, c_t = 3) \\ 0 & \text{otherwise} \end{cases}$, where $t$ indicates the trial index,

$N_t(S3)$ indicates the number of visiting $S3$ at trial $t$, and $c_t$ indicates the hidden state at trial $t$.

- **Hidden state** $c_t = \begin{cases} 2 & \text{if } \mathrm{mod}(t, 60) \leq 30 \\ 3 & \text{otherwise} \end{cases}$, where hidden variable switches depending on the

trial index.

## Simplified probabilistic cueing task (**Figures 4 and 5**)

Simplified probabilistic cueing task (**Figures 4 and 5**) can be described as follows.

- **State space** $S = \{S_0, S_1, S_2, S_3, S_4, S_5\}$, where $S_0$ ($if\ c_t = 1$) or $S_1$ ($if\ c_t = 2$) are the starting points and $S_4$ and $S_5$ are the reward delivery points.
- **Action space** $A = \{a_{0(23)}, a_{1(23)}, a_{24}, a_{25}, a_{34}, a_{35}\}$, where each action determines a state transition.

- **Transition function** $T(s_j \mid s_i, a_{ij}) = \begin{cases} 1-p & \text{if } (i,j) \in \{(0,3),(1,2)\} \\ p & \text{if } (i,j) \in \{(0,2),(1,3)\} \\ 1 & \text{otherwise} \end{cases}$, specifying the probability of

reaching state $S_j$ given current state $S_i$ and action $a_{ij}$. We set p=0.8 in **Figure 4**, and we set

p=0.5 in **Figure 5**.

- **Reward function** $R_t(s, a, c_t) = \begin{cases} 1 & \text{if } (s = S_4, a = a_{24}, c_t = 1), \\ 1 & \text{if } (s = S_5, a = a_{35}, c_t = 1), \\ 1 & \text{if } (s = S_4, a = a_{34}, c_t = 2), \\ 1 & \text{if } (s = S_5, a = a_{25}, c_t = 2), \\ 0 & \text{otherwise.} \end{cases}$, where $t$ indicates the trial index,

and $c_t$ indicates the hidden state at trial $t$.

- **Hidden state** $c_t = \begin{cases} 1 & \text{if } \mathrm{mod}(t, 40) \leq 20, \\ 2 & \text{otherwise.} \end{cases}$, where hidden variable switches depending on the

trial index.

## Model-free learning with temporal contexts

To highlight the advantage of our model, we compared it to the Q-learning with temporal contexts (**Figure 3—figure supplement 1**), namely, the state is defined by the recent $n$-step history of environmental state (i.e. $s_k^{(n)} = (S_k, S_{k-1}, \cdots, S_{k-n})^T$, where $s_k^{(n)}$ is the temporal context state, and $S_k$ is the environmental state at time $k$). We changed $n$ from 0 to 3. In the Q-learning, the action value for a temporal state $s_k$ to the next $s_{k+1}$ is updated as

$$Q\left(s_k, s_{k+1}\right) \leftarrow \left(1 - \alpha\right) Q\left(s_k, s_{k+1}\right) + \alpha \left(R\left(s_{1:\,k+1}\right) + \gamma \max_s Q\left(s_{k+1}, s\right)\right) \tag{17}$$

where the initial $Q$ value is 0, learning rate $\alpha$=0.4, the discount factor $\gamma$=0.6, and the task-dependent reward function $R$=100 for the rewarded transition and $R$=1 for else. Next, state selection policy $\pi$ is set to be proportional to $Q$ value as

$$\pi\left(s_k, s_{k+1}\right) \propto Q\left(s_k, s_{k+1}\right) \tag{18}$$

## Inhibition experiment

To replicate the inhibition experiment of medial EC axons at CA1, we inhibit 98.5% of the input from the context domain of $X$ to $H$. After the 2-lap task in **Figure 3**, we observed the hippocampal activity responding to each contextual state with or without this inhibition. ESR correlation is calculated based on the hippocampal activity of each lap, while the spatial correlation is calculated based on that of space. To avoid NaN value when calculating correlations, we assumed that the activity of hippocampal cells without firing would have a random spontaneous activity between 0 and 0.1. Note that this operation does not significantly affect the result.

## Acknowledgements

The study was supported by RIKEN Center for Brain Science, the JST CREST program JPMJCR23N2, JSPS KAKENHI 25K24466, and RIKEN TRIP initiative (RIKEN Quantum).

## Additional information

### Funding

| Funder | Grant reference number | Author |
| --- | --- | --- |
| Japan Science and Technology Agency | 10.52926/jpmjcr23n2 | Taro Toyoizumi |
| RIKEN | TRIP initiative: Quantum | Taro Toyoizumi |
| Japan Society for the Promotion of Science | KAKENHI 25K24466 | Taro Toyoizumi |

The funders had no role in study design, data collection and interpretation, or the decision to submit the work for publication.

### Author contributions

Yoshiki Ito, Conceptualization, Resources, Data curation, Software, Formal analysis, Validation, Investigation, Visualization, Methodology, Writing – original draft, Writing – review and editing; Taro Toyoizumi, Conceptualization, Supervision, Funding acquisition, Methodology, Writing – original draft, Project administration, Writing – review and editing

### Author ORCIDs

Yoshiki Ito ⓘ https://orcid.org/0000-0002-4867-726X
Taro Toyoizumi ⓘ https://orcid.org/0000-0001-5444-8829

Reviewer #2 (Public review): https://doi.org/10.7554/eLife.106506.4.sa1
Reviewer #3 (Public review): https://doi.org/10.7554/eLife.106506.4.sa2

Author response https://doi.org/10.7554/eLife.106506.4.sa3

## Additional files

**Supplementary files**
MDAR checklist

**Data availability**
All data needed to evaluate the conclusions in the paper are present in the paper and/or the Supplementary Materials. All source code is provided in https://github.com/toppo365/flexiblemodel (copy archived at *Ito, 2026*).

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
