## [Editor Report · eLife Assessment]

This manuscript reports a **valuable** modeling study on sequence generation in the hippocampus in a variety of behavioral contexts. The authors model context-depending decision making, and suggest that psychiatric disorders can be interpreted in terms of over or under representation of context information. The presentation is **solid**, and the work will interest the broad community of researchers studying cortical-hippocampal interactions and sequences.

---

## [Referee Report · Reviewer #2 (Public review)]

[Editors' note: This version has been assessed by the Reviewing Editor without further input from the original reviewers. The authors have addressed the comments raised in the previous round of review.]

Summary:

Ito and Toyoizumi present a computational model of context-dependent action selection. They propose a "hippocampus" network that learns sequences based on which the agent chooses actions. The hippocampus network receives both stimulus and context information from an attractor network that learns new contexts based on experience. The model is consistent with a variety of experiments both from the rodent and the human literature such as splitter cells, lap cells, the dependence of sequence expression on behavioral statistics. Moreover, the authors suggest that psychiatric disorders can be interpreted in terms of over/under representation of context information.

My general assessment of the work is unchanged, and I still have some questions requesting methodological clarification

Strengths:

This ambitious work links diverse physiological and behavioral findings into a self-organizing neural network framework. All functional aspects of the network arise from plastic synaptic connections: Sequences, contexts, action selection. The model also nicely links ideas from reinforcement learning to a neuronally interpretable mechanisms, e.g. learning a value function from hippocampal activity.

---

## [Referee Report · Reviewer #3 (Public review)]

Summary:

This paper develops a model to account for flexible and context-dependent behaviors, such as where the same input must generate different responses or representations depending on context. The approach is anchored in the hippocampal place cell literature. The model consists of a module X, which represents context, and a module H (hippocampus), which generates "sequences". X is a binary attractor RNN, and H appears to be a discrete binary network, which is called recurrent but seems to operate primarily in a feedforward mode. H has two types of units (those that are directly activated by context, and transition/sequence units). An input from X drives a winner-take-all activation of a single unit H_context unit, which can trigger a sequence in the H_transition units. When a new/unpredicted context arises, a new stable context in X is generated, which in turn can trigger a new sequence in H. The authors use this model to account for some experimental findings, and on a more speculative note, propose to capture key aspects of contextual processing associated with schizophrenia and autism.

Strengths:

Context-dependency is an important problem. And for this reason, there are many papers that address context-dependency - some of this work is cited. To the best of my knowledge, the approach of using an attractor network to represent and detect changes in context is novel and potentially valuable.

---

## [Author Response]

The following is the authors’ response to the previous reviews

**Public Reviews:**

**Reviewer #2 (Public review):**
Summary:Ito and Toyoizumi present a computational model of context-dependent action selection. They propose a "hippocampus" network that learns sequences based on which the agent chooses actions. The hippocampus network receives both stimulus and context information from an attractor network that learns new contexts based on experience. The model is consistent with a variety of experiments both from the rodent and the human literature such as splitter cells, lap cells, the dependence of sequence expression on behavioral statistics. Moreover, the authors suggest that psychiatric disorders can be interpreted in terms of over/under representation of context information.My general assessment of the work is unchanged, and I still have some questions requesting methodological clarificationStrengths:This ambitious work links diverse physiological and behavioral findings into a self-organizing neural network framework. All functional aspects of the network arise from plastic synaptic connections: Sequences, contexts, action selection. The model also nicely links ideas from reinforcement learning to a neuronally interpretable mechanisms, e.g. learning a value function from hippocampal activity.Weaknesses:The presentation, particularly of the methodological aspects, needs to be heavily improved. Judgment of generality and plausibility of the results is severely hampered but is essential, particularly for the conclusions related to psychiatric disorders. In its present form, it is impossible to judge whether the claims and conclusions made are justified. Also, the lack of clarity strongly reduces the impact of the work on the field.

Thank you for pointing this out.

In the revised text, we clarified the definition of “time step” and how hippocampal neurons behaved in each time step (see individual comments below). Also, we clarified the implementation of disorder conditions in our model by indicating the exact neuron numbers of the stimulus domain in H module as below. (Other parameters were common in all conditions.)

“𝑋 consists of two domains: stimulus domain 𝑋 and context domain 𝑋. The neuron ratio in the stimulus domain over the whole neurons dim 𝑋/𝑁 is 16.7% (200 neurons) for the control condition, 2.5% (30 neurons) for the SZ condition, and 50% (600 neurons) for the ASD condition.”

Comments:The authors have made strong efforts to improve on their description of the methods, however, it is still very hard to understand. As a result of some of their clarifications, new issues appeared that I was not able to extract in the previous version.(1) Particularly I had problems figuring out how the individual dynamical systems are interrelated (sequences, attractor, action, learning). As I understand it now (and I still might be wrong) there is one discrete time dynamics, where in each time step one action takes place as well as the attractor and sequence dynamics are moved one step forward. Also, synaptic updates happen in every one of those time steps. The authors may verify or correct my interpretations and further improve on their description in the manuscript. It is also confusing that time in the figure panels is given in units of trials, where each trial may consist of (maybe different amounts of) multiple time steps. Are the thin horizontal red ad blue lines time steps?

Thank you for raising the confusing point.

The reviewer’s understanding is correct. In our model, at each time step the agents transition to the next environmental state (which also corresponds to the contextual state). During this step, each processing stage proceeds in order: Context selector performs attractor selection, Sequence composer performs sequence selection, followed by action selection and synaptic updates. As learning progresses and hippocampal sequences begin to predict longer futures, reducing the need for step-by-step planning. However, at least at the beginning of each task, all processes are conducted at each time step (see Fig. 1G).

In all tasks, trials are reset when the agents visit the reward sites (i.e., S4 or S5). n Fig. 2C, for example, one trial consists of three time steps (i.e., three state transitions), and the red and blue shaded regions indicate individual trials. During each time step, two types of hippocampal neurons are activated: a state-coding neuron and a transition-coding neuron. (In contrast, in X, one contextual state is active during one time step). Therefore, in Fig. 2E, two neuronal activities correspond to a single time step.

For clarification, we have revised Fig. 2 and related descriptions in the manuscript as follows.

“Here, we simplified this task by using an environment with five discrete states (S1-S5), i.e., five discrete external stimuli (Figure 2A), where agents transition to the next state at each time step.”

“Figure 2C illustrates an example of both the environmental state transition and the corresponding contextual state transition of an agent, with each trial resetting upon visiting the reward sites (S4 or S5). ”

“At each time step, one state-coding neuron and one transition-coding neuron are active in this order.”

“At each time step, the agents transition between environmental states.”

“The model’s computational dynamics are fundamentally synchronized with the environmental (behavioral) time step, and at each time step, the agents transition to the next environmental state. Upon a state transition, the agents first perform contextual state estimation by Context selector and activate a corresponding hippocampal neuron.”

(2) As a consequence of my new understanding of the model dynamics, I have become doubts about the interpretation of the attractor network as context encoding. Since the X population mainly serves to disambiguate sequence continuation, right before the action has to be taken (active for only two time steps in Figure 1C?) they could also be considered to encode task space (El-Gaby et al. 2024; doi: 10.1038/s41586-024-08145-x).

We thank the reviewer for this insightful comment.

First of all, we would like to clarify that Figure 1C shows the following process: the activity of H at time step t−1 and the external stimulus at time step t jointly provide input to X module, and the activity of X settles into a contextual state at the time step t. As explained in our response to comment (1), the activity of X remains constant during each time step.

The primary function of X module in our model is to disambiguate the environmental states defined by the external stimuli based on the history information. It is true that, in practice, whether an ongoing sequence is maintained or remapped depends on whether the observed stimulus is consistent with the predicted stimulus. However, this is a consequence of the predictive sequence obtained from scratch rather than the primary computational role of X module. In contrast, X module becomes particularly important when past experience does not uniquely determine the next state. In this situation, the agent must infer the contextual state by associating the current situation with previously experienced contexts, rather than relying solely on temporal continuity.

We also add that, in most successful cases, the contextual states learned by the agent often correspond to the hidden states of each task as a result of disambiguation. In this sense, the resulting representation may resemble a “task space” encoding, as suggested by the reviewer. However, an important aspect of our model is that the agent does not assume the existence or number of hidden states a priori. Instead, we considered the situation where the agent initially underestimates the number of contextual states, and through remapping it incrementally increases the number of contextual representations. When the number of contextual states matches the number of hidden task states, the task is typically solved.

(3) Also technically, I wonder why the authors introduce the criterion of 50(!) time steps to allow the attractor to converge, if the state of the attractor network is only relevant in one time step to choose the appropriate continuation of the sequence of actions. Is attractor dynamics important at all? What would happen if just the input and output weights to the X population are kept and the recurrent weights are set 0?

We thank the reviewer for raising this confusing point.

First, we would like to clarify that the “50 iterations” mentioned in the manuscript does not refer to 50 environmental time steps. We implemented multiple iterations of attractor updates (typically until convergence) by Context selector within each behavioral time step.

We clarify this point in the Method section as below.

“After history-based or landmark-based initialization, X iteratively updates its contextual state at the beginning of each time step according to the associative memory dynamics:”

The recurrent connectivity within the X population is essential for attractor updates. If the recurrent weights were removed (i.e., set to zero), the network would lose the ability to retrieve distinct contextual states for the same stimulus. In that case, the model would be unable to solve the context-dependent task as we showed in this manuscript.

(4) Figure 3E: How many time steps are the H cells active (red bars?) Figure 4J: What are the units of the time axis?

Thank you for pointing this out.

In Figure 3E, each time step is indicated in the X-axis ticks (i.e., each environmental state). As we explained in the comment (1), two hippocampal neurons’ activity (red bars) corresponds to each time step.

Similarly, in Figure 4J, each time step is indicated in the X-axis ticks. To better represent the results, we added descriptions of the environmental states in our model to the X-axis tick labels in Figure 4J.

We added the following texts below in Figure captions.

“The x-axis represents each time step (corresponding to environmental states), and the y-axis shows the sorted activity of H module.”

“The x-axis represents each time step (corresponding to environmental states), and the y-axis shows the decoding accuracy of each context based on hippocampal activity.”